# Retrieval of sea ice drift in the Fram Strait based on data from Chinese satellite HaiYang (HY1-D CE1)

**Dunwang Lu**[1,2,3], **Jianqiang Liu**[2,3,✝], **Lijian Shi**[2,3], **Tao Zeng**[2,3], **Bin Cheng**[4], **Suhui Wu**[2,3], **and Manman Wang**[1,2,3]

[1]National Marine Environmental Forecasting Center, Beijing 100081, China
[2]National Satellite Ocean Application Service, Beijing 100081, China
[3]Key Laboratory of Space Ocean Remote Sensing and Application, Ministry of Natural Resources, Beijing 100081, China
[4]Finnish Meteorological Institute, Helsinki 00101, Finland
✝deceased

**Correspondence:** Lijian Shi (shilj@mail.nsoas.org.cn)

**Abstract.** TS1 Melting of sea ice in the Arctic has accelerated due to global warming. The Fram Strait (FS) serves as a crucial pathway for sea ice export from the Arctic to the North Atlantic Ocean. Monitoring sea ice drift (SID) in the FS provides insight into how Arctic sea ice responds to the climate change. The SID has been retrieved from Sentinel-1 synthetic aperture radar (SAR), Advanced Very High Resolution Radiometer (AVHRR), Moderate Resolution Imaging Spectroradiometer (MODIS), and Advanced Microwave Scanning Radiometer for EOS (AMSR-E), and further exploration is needed for the retrieval of SID using optical imagery. In this paper, we retrieve SID in the FS using the Chinese HaiYang1-D (HY1-D) satellite equipped with the Coastal Zone Imager (CZI). A multi-template matching technique is employed to calculate the cross-correlation, and subpixel estimation is used to locate displacement vectors from the cross-correlation matrix. The dataset covering March to May 2021 was divided into hourly and daily intervals for analysis, and validation was performed using Copernicus Marine Environment Monitoring Service (CMEMS) SAR-based product and International Arctic Buoy Programme (IABP) buoy. A comparison with the CMEMS SID product revealed a high correlation with the daily interval dataset; however, due to the spatial and temporal variability in the sea ice motion, differences are observed with the hourly interval dataset. Additionally, validation with an IABP buoy yielded a velocity bias of $-0.005\,\mathrm{m\,s^{-1}}$ TS2 and RMSE of $0.031\,\mathrm{m\,s^{-1}}$ for the daily interval dataset, along with a flow direction bias of $0.002\,\mathrm{rad}$ and RMSE of $0.009\,\mathrm{rad}$, respectively. For the hourly interval dataset, the velocity bias was negligible ($0\,\mathrm{m\,s^{-1}}$), with a RMSE of $0.036\,\mathrm{m\,s^{-1}}$, while the flow direction bias was $0.003\,\mathrm{rad}$, with a RMSE of $0.010\,\mathrm{rad}$. In addition, during the validation with buoys, we found that the accuracy of retrieving the SID flow direction is distinctly interrelated with the sea ice displacement.

## 1 Introduction

The Arctic, as one of the three poles of the Earth (Li et al., 2020) that stores $101\,000\,\mathrm{km^3}$ of fresh water (Finnish Meteorological Institute et al., 2022), is an important part of the cryosphere and plays a prominent role in global water resources and atmospheric cycles. Since the beginning of the 21st century, global warming has profoundly affected human production and activities and has become a strong threat to the stability of the climate system (Cook et al., 2014). Under the effect of Arctic amplification (Serreze et al., 2009), Arctic quick warming accelerates the melting of polar sea ice, leading to thinner sea ice and accelerating sea ice transport (Krumpen et al., 2016; Maslanik et al., 2011). The acceleration of sea ice motion also indicates a reduction in the sea ice residence time in the Arctic (Sumata et al., 2023). Moreover, changes in sea ice feed back into the climate system, affecting energy transport (Dethloff et al., 2006; Döscher et al., 2014).

Sea ice drift (SID) is an important geophysical parameter to describe the dynamic of sea ice and the sea ice motion under the influence of winds, currents, and various external forces (gravity, Coriolis effect, etc.) (Encyclopedia of Ocean Sciences, 2022). The primary SID circulation across the Arctic encompasses both the Beaufort Gyre (BG) and the transpolar drift (TPD) (Preller and Posey, 1989), and the TPD transports large quantities of multiyear ice outward from the central Arctic toward the FS, Barents Sea, and Baffin Bay (Colony and Thorndike, 1984; Martin and Augstein, 2000). The FS connects the Arctic and the North Atlantic (Sumata et al., 2022), and large quantities of sea ice are injected into the North Atlantic each year through the strait (Reimnitz et al., 1994). The long-term annual average ice outflow in the FS (1935–2014) is approximately $880\,000\,\text{km}^3$, accounting for 10 % of the sea ice cover in the Arctic Basin (Smedsrud et al., 2017) and the largest portion (90 %) of the Arctic sea ice export volume (Sumata et al., 2022; Haine et al., 2015; Serreze et al., 2006). Sea ice is a mixture of ice and brine (Schwerdtfecer, 1963), which gradually disintegrates during outward transport in the FS. This process affects freshwater exchange and energy transport in the North Atlantic, which may alter the convective overturning of water masses and thermohaline circulation processes (Aagaard and Carmack, 1989). In addition, for shipping planning and scientific research, the progression of polynyas and lead is heavily influenced by SID (Wagner et al., 2021). Therefore, observing SID in the FS is crucial to analyze the sea ice variation in the Arctic and the sea ice transport from the polar to subpolar regions.

The development of satellites and remote sensing sensors promote satellite data as a prevailing trend in retrieving SID (Kwok, 2010). The SID retrieved from satellite-derived data possesses spatiotemporal continuity, which helps researchers better evaluate the sea ice variation in the Arctic and provides a basis for climate forecasting and ship route planning. At present, the primary data used in SID retrieval are radiometer, scatterometer, synthetic aperture radar (SAR), and optical imagery. SID products derived from radiometers and scatterometers inherently possess a coarse spatial resolution, owing to the characteristics of the sensors. Low-resolution SID products exhibit large error in the velocity in the FS (Hwang, 2013). The Ocean and Sea Ice Satellite Application Facility (OSI SAF) provides SID products retrieved from scatterometers and radiometers over the polar regions and its temporal coverage is from 2009 to now; however, the time interval of these products is greater than 1 d. Due to the complexity of sea ice dynamics, a low-temporal-resolution SID product may fail to depict the subdaily scale variation in the sea ice motion. High-spatiotemporal-resolution SID products are valuable for capturing subtle SID patterns (Johansson and Berg, 2016). The SAR image provides high spatial resolution, but radar backscatter is sensitive to the liquid water on sea ice surface, and the accuracy of retrieving SID during the melting season can be hampered (Stern and Moritz, 2002). This disadvantage constrains the application of SAR in retrieving SID. Optical imagery has been applied extensively in cryosphere observation. Many in-orbit satellites equipped with optical imagery and optical imagery have the advantages of a wide swath and high resolution, which are beneficial for capturing the surface roughness and textural characteristics of sea ice. Optical imagery has the potential to retrieve SID in the FS. While optical imagery has the advantages of high resolution and wide swath, this can also significantly increase the computational effort of the algorithm. To solve this problem, we applied the MCC (maximum cross-correlation) algorithm with improvements for SID retrieval. The MCC has been widely used in retrieving SID with radiometer and scatterometer, as well as with SAR (Girard-Ardhuin and Ezraty, 2012; Hollands and Dierking, 2011). However, when it comes to utilizing optical imagery for SID retrieval, many researchers tend to use different algorithms instead of the MCC. Petrou and Tian (2017) `TS3` designed an algorithm based on optical flow to retrieve SID from Moderate Resolution Imaging Spectroradiometer (MODIS). However, due to the absence of buoys in their study area, the accuracy of their algorithm is unevaluated. Lopez-Acosta et al. (2019) `TS4` developed a complex ice-tracking algorithm specifically designed for retrieving SID in the FS but requiring multiple MODIS images to generate a complete SID field. Fang et al. (2023) `TS5` discussed the potential of feature tracking with MODIS images for SID retrieval. The accuracy of their result is promising, but the spatial coverage needs to be further improved. Wang et al. (2021) designed an ice-tracking algorithm to retrieve SID in the marginal ice zone (MIZ), but the algorithm cannot produce an intact SID field. The European Organization for the Exploitation of Meteorological Satellites (EUMETSAT) uses the MCC algorithm to retrieve SID from Advanced Very High Resolution Radiometer (AVHRR), with a spatial resolution of 20 km and a temporal resolution of 1 d (Dybkjaer, 2018). However, it has been found that the accuracy of the SID product retrieved from AVHRR presents low accuracy in eastern Greenland, with the mean absolute error (MAE) of velocity reaching $10.40\,\text{km}\,\text{d}^{-1}$, which is even lower than that of the SID products retrieved from radiometer and scatterometer (Wang et al., 2022). In our study, we applied multi-template matching combined with a subpixel estimation for SID retrieval. The method of multi-template matching achieves a balance between computational efficiency and accuracy, and the subpixel estimation further improves the accuracy of the retrieval results while saving computational resources. We believe that the MCC algorithm with improvements can be utilized for retrieving SID effectively from optical imagery.

This paper focuses on designing an improved MCC algorithm using the Coastal Zone Imager (CZI) of HaiYang-1D (HY-1D) data to retrieve SID in the FS. Moreover, the retrieved results are evaluated via comparison with SAR-based SID products released by Copernicus Marine Environment Monitoring Service (CMEMS) and International Arctic Buoy Programme (IABP) buoy data. For sea ice motion, the av-

erage velocity of the sea ice motion in Arctic is $0.02\,\mathrm{m\,s^{-1}}$ (Colony and Thorndike, 1984), and the maximum velocity of the sea ice in the FS is $0.64\,\mathrm{m\,s^{-1}}$ (Lei et al., 2016). The multiyear ice (MYI) drifts from the Arctic Basin and crushes in the central part of strait, which results in the fragmented ice in the southern part of the FS and along the eastern coast of Greenland. The fragmented ice drifts at a faster velocity near the FS compared to sea ice within the Arctic Basin. The sea ice characteristics described above increase the difficulty of retrieving SID in the FS.

This paper is structured as follows. Section 2 describes the data used in this study, including the CZI image, CMEMS SID products, and IABP buoy. Section 3 details the algorithm used to retrieve SID. Section 4 compares the results with CMEMS SID products and buoy data for validation. Section 5 discusses the retrieved results. Finally, concluding remarks are provided in Sect. 6.

## 2 Data

### 2.1 HY-1D CZI data

HaiYang-1D (HY-1D) satellite was launched in 11 June 2020. The satellite is equipped with the Chinese Ocean Color and Temperature Scanner (COCTS), CZI, UltraViolet Imager (UVI), satellite calibration spectrum, and automatic identification system. The revisit period of the CZI is 3 d, the resolution of the CZI image is 50 m, and the swath of the CZI image is nearly 950 km. CZI possesses four bands (visible near-infrared, VIR, and near-infrared, NIR), as shown in Table 1. The wide swath and high spatial resolution of the CZI imagery make it suitable for the sea ice motion observation in the FS.

Since the launch of China's first ocean satellite in 2002, concerted efforts have been undertaken to institute a comprehensive global operational ocean satellite observation system. Currently, the observation system consists of 10 satellites, which include three series: ocean color series satellites (HY-1), ocean dynamic environment series satellites (HY-2), and ocean surveillance and monitoring series satellites (HY-3) (Zeng et al., 2023). In this paper, SID retrieval is performed using the L1C data, which are processed with radiometric calibration and geographic projection. Before retrieving, the CZI images are resampled to 300 m, considering the algorithm's computational efficiency and the spatial resolution of the result.

In the FS, the climate condition is harsh, and sea ice changes rapidly in response to wind (Tsukernik et al., 2010), which is a potentially hazardous factor for shipping and scientific expeditions, achieving timely and accurate monitoring of SID is a crucial goal. The CZI image has a high spatial resolution and wide swath for providing accurate monitoring of the SID. In our study, 111 CZI images from March to May 2021 were used to produce the intersection region

**Table 1.** Information on the HY-1D CZI data.

| Band | Wavelength (µm) | Band name |
|------|-----------------|-----------|
| 1 | 0.42–0.50 | Blue |
| 2 | 0.52–0.60 | Green |
| 3 | 0.61–0.69 | Red |
| 4 | 0.76–0.89 | Near-infrared |

(as shown in Fig. 1a) for the dataset, and 48 pairs of images were eventually produced for SID retrieval. Furthermore, to explore the influence of the image time interval on retrieval SID, the 48 pairs of image datasets are divided into a dataset with 16 pairs of hour level CE2 (less than 6 h) data and a dataset with 32 pairs of day level (approximately 24 h) data. Figure 1b shows the time intervals of image pairs, and Fig. 1c plots the outer frame lines of the image pairs, where the white borders are the locations of the image pairs with an hour level interval, and the darker blue borders are the locations of the image pairs with a day level interval.

### 2.2 CMEMS SID product

As the CZI images possess a high spatial resolution, the SID products retrieved from the scatterometer and radiometer are not comparable for validation. Thus, the SAR-based SID product is chosen for comparison. The Global Ocean – High Resolution SAR Sea Ice Drift is a polar near-real time gridded sea ice drift product produced by the National Space Institute at the Technical University of Denmark (DTU space) (European Union–Copernicus Marine Service, 2015). The data are released by the Copernicus Marine Environment Monitoring Service (CMEMS) and distributed on the website of Copernicus Marine Service. The composite product is updated every 12 h and covers 24 h. Thus, two kinds of product are provided with a nominal time intervals of 00:00 to 00:00 and 12:00 to 12:00 TS6, respectively. The grid resolution of the product is 10 km, and the product is available from both Arctic Ocean and Southern Ocean (Pedersen et al., 2015). The product has been validated using Woods Hole ice-tethered profilers (ITPs). For the validation of 2021, the number of matched pairs is 29 180, the correlation coefficient between the product and ITP buoys is 0.99, and root mean square deviation (RMSD) of d$x$ and d$y$ is 362.32 m ($0.0042\,\mathrm{m\,s^{-1}}$) and 339.81 m ($0.0039\,\mathrm{m\,s^{-1}}$), the bias of d$x$ and d$y$ is 4.64 and 17.29 m, and the bias of velocity is negligible ($\sim 0\,\mathrm{m\,s^{-1}}$). The validation is performed with the 24 h mean composite product. (European Union–Copernicus Marine Service, 2015). Sea ice motion changes rapidly in the FS, and the SID results retrieved from images at different times are incompatible. Therefore, the CMEMS SID product which has the most temporal overlap with CZI images is chosen for comparison. Additionally, due to the disparity in spatial resolution, we resampled the retrieved SID to the resolution of the CMEMS SID product by linear interpolation.

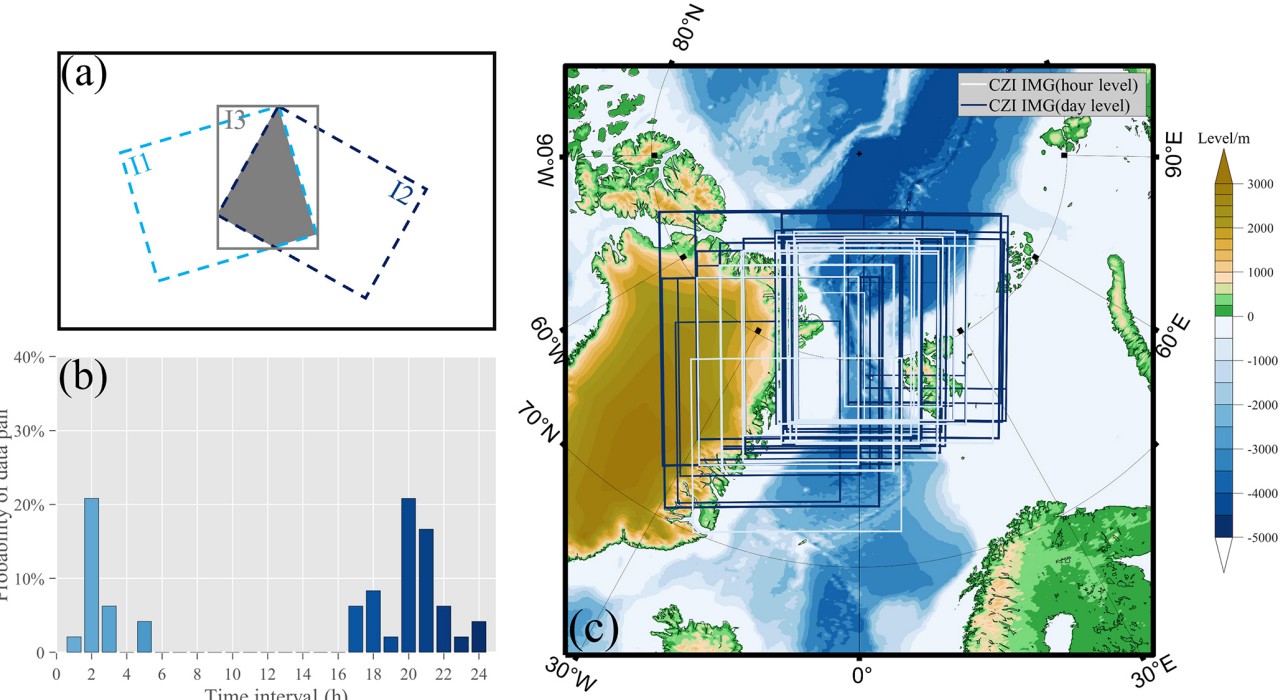

**Figure 1.** An illustration showing the creation of the dataset using CZI images **(a)**. The time interval histogram of the dataset **(b)** and the location of the image pairs **(c)**.

## 2.3 IABP buoys

GPS position data from buoys are the most credible data for the retrieved SID validation. The International Arctic Buoy Programme (IABP) has deployed a network of drifting buoys in the Arctic to provide data for scientific research (Rigor et al., 2008). IABP has collected more than 500 months of data from more than 500 buoys deployed in the Arctic. Buoy data collected from IABP are an essential component of Arctic scientific data. These data have been used to validate retrieved SID (Hakkinen et al., 2008; Fang et al., 2023). The drift trajectories of the 50 buoys used in our study are shown in Fig. 2, and these drift trajectories consist of consecutive GPS points. Among the buoys used for validation, 31 buoys from the Multidisciplinary drifting Observatory for the Study of Arctic Climate (MOSAiC) were included. The GPS position of MOSAiC buoys has an accuracy of $\pm 2.5\,\mathrm{m}$ (Qiu and Li, 2022), and the IABP buoy has an accuracy of approximately $300\,\mathrm{m}$ (Haarpaintner, 2006). We set a minimal distance of 4 km to search the buoys for validation. Because some of the buoys were deployed in the vicinity, we used the average of them to validate the result. Ultimately, 206 matched drift vectors were produced for validation.

## 3 Improved SID retrieval method in the Fram Strait

The flowchart of our study is shown in Fig. 3. We choose same orbit images of different moments to calculate the intersection, and the intersection area is clipped to generate dataset. Data enhancement in preprocessing is used to distinguish between sea ice and seawater and highlight the textural details of the sea ice. Data enhancement is practical for retrieving SID using optical imagery (Fang et al., 2023; Lopez-Acosta et al., 2019; Yan et al., 2023). After preprocessing, the sea ice displacement will be retrieved via multi-template matching and subpixel estimation. Since the optical images are susceptible to various factors, such as cloud cover and the angle of incidence of the light (Stow et al., 2004), the SID contains noise and low-quality results. We designed the quality control session to remove the low-quality results and noise from the SID. The penultimate step involves smoothing and filling to derive an intact SID field. Finally, we validate the results using buoys and the CMEMS SID product.

## 3.1 Data enhancement

Before preprocessing, the resampled image radiance values are converted to grayscale (0–255). Then, the data of different bands are stretched logarithmically. After reviewing study in which optical imagery was used to retrieve SID (Fang et al., 2023), we selected the average of the bands 2, 3, and 4 as the input data. Finally, histogram equalization and

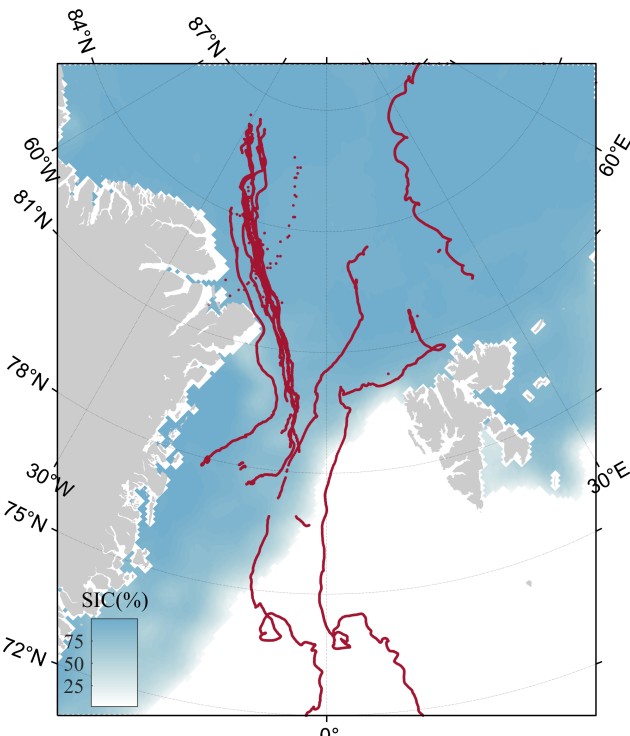

**Figure 2.** Drifting trajectories of 50 IABP buoys from March to May 2021, with the percent of SIC (sea ice concentration) also shown.

edge detection are applied to the image. The process and effect of data enhancement are explicitly described as follows.

In the polar regions, the lighting conditions are complex, and some images are captured at a slight solar zenith angle (Stow et al., 2004). These disadvantages could lead to an uneven spatial distribution of image gray values, which affects the retrieval of the SID. Logarithmic stretching is an image grayscale transformation that expands the low gray value portion and compresses the high gray value portion (Lupton et al., 2004). Equation (1) is the formula for logarithmic stretching, where $r$ is the gray value of the image, and $c$ is the base. The larger the base is, the stronger the emphasis on the lower gray portions and the stronger the compression of the higher gray portions. As shown in Fig. 4a and b, logarithmic stretching improves the uneven lighting and enhances the sea ice texture.

$$T(r) = c \cdot \log(1 + r) \tag{1}$$

In addition to logarithmic stretching, histogram equalization (HE) is also useful for adjusting the grayscale distribution of image (Krutsch and Tenorio, 2011). The adaptive histogram equalization (AHE) algorithm is a variant of histogram equalization (Pizer et al., 1987). The contrast is changed by calculating the local histogram of the image and then redistributing the value. The algorithm is an improvement over the traditional HE, but the disadvantage

of AHE is that it introduces noise during grayscale stretching. Currently, contrast-limited adaptive histogram equalization (CLAHE) has improved upon AHE (Zuiderveld, 1994). Therefore, we chose CLAHE to redistribute the gray values. The image histograms during grayscale transformation are shown in Fig. 4d. Logarithmic stretching improves the gray value of the image (Fig. 4b), and CLAHE enhances the texture and gap features in high sea ice concentration regions (Fig. 4c).

Cloud cover can lead to loss of contrast and diminished textural detail, affecting the quality of the result. Edge detection, the detection and characterization of significant intensity changes (Torre and Poggio, 1986), is valuable for reinforcing the texture differences between cloud cover and sea ice. In our study, Sobel edge detection is applied to enhance the texture, and we find that edge detection can also neutralize some disadvantageous effect of cloud cover. We will describe this aspect thoroughly in the discussion section.

## 3.2 Sea ice drift retrieval

### 3.2.1 Multi-template matching

The traditional template matching requires searching in the vicinity of the template to determine the location with the largest correlation coefficient, which consumes considerable time. Multi-template matching is performed with different template sizes, ranging from coarse resolution to fine resolution, reducing calculation time while ensuring accuracy (Wang et al., 2008). As shown in Fig. 5, the time cost of multi-template matching for the 48 datasets is less than 150 s, and the average number of image pixels used for the retrieval is 5 208 298. This indicates that multi-template matching has high computational efficiency for wide-swathed images. In addition, cross-correlation calculation is required during template matching, and the calculation is slow in the spatial domain (Thielicke and Stamhuis, 2014) and cannot yield a correlation matrix with equal size to the template. Therefore, we calculate the correlation in the frequency domain using fast Fourier transform (FFT).

### 3.2.2 Subpixel estimation

Identifying the peak of the correlation coefficient is an essential factor for improving the accuracy of retrieval. The integer displacements of the two templates can be determined directly from the peak positions of the correlation matrix. However, this approach introduces quantization error, and the difference in the minimum displacements will be multiples of the image resolution (Lavergne et al., 2010). To address this issue, subpixel precision refinement of the peak position with the biquadratic surface fitting method has been proposed to mitigate quantization errors (Kwok et al., 1998). Subpixel estimation could result in considerable computational costs when using sophisticated fitting (Hollands and

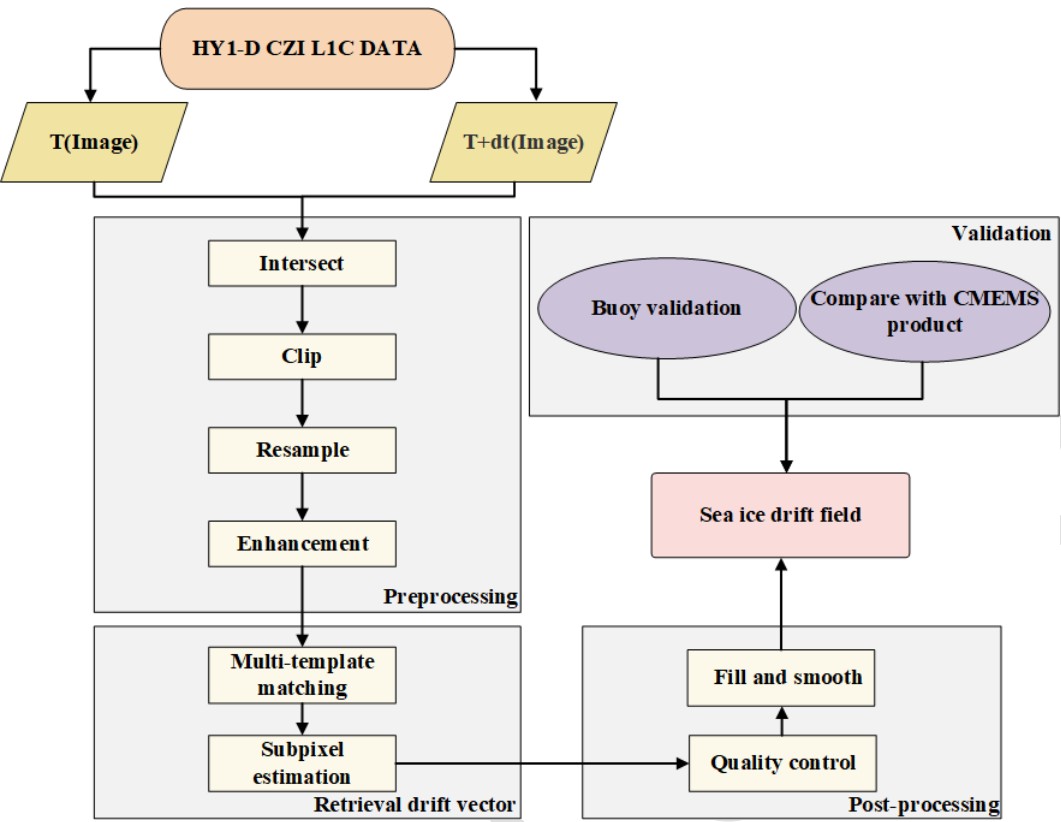

**Figure 3.** Flow chart of sea ice drift retrieval.

Dierking, 2011). Therefore, for our study, we have opted for simple Gaussian fitting, since this approach strikes a balance between computational efficiency and accuracy (Thielicke and Stamhuis, 2014).

### 3.3    Quality control

The retrieved SID obtained through multi-template matching and subpixel estimation is presented in Fig. 6a. However, due to the influence of cloud cover and seawater, certain areas of SID have a low confidence level. The algorithm may extract erroneous SID vectors, and it is necessary to filter those erroneous vectors. Therefore, we employ a series of methods to control the quality of the SID. By applying quality control, consistency within the SID is ensured while eliminating erroneous drift vectors. Subsequent sections provide detailed descriptions of the quality control procedures.

#### 3.3.1    Cross-correlation value and its derived parameter filter

The magnitude of the cross-correlation ($R$) in the MCC serves as an indicator of the quality of the result (Haarpaintner, 2006). It represents the correlation between two templates, making it an effective parameter for the initial quality control. Additionally, we employ two parameters derived

from the cross-correlation matrix to further control the quality. In the selection of the peak, incorrect peaks which are observed near the peak can be labeled as subpeaks. The ratio of peak to subpeak was used to measure of the "uniqueness" of the real peak. Furthermore, comparing the average of the correlation coefficient matrix with the peak provides a measure of the signal-to-noise level of the peaks. Two parameters, the peak mean ratio (PMR) (Eq. 2) and the peak second ratio (PSR) (Eq. 3) (Dybkjaer, 2018; Van Wyk de Vries and Wickert, 2021), are used to measure the quality of the result.

$$\mathrm{PMR} = \frac{C_{\mathrm{peak}}}{\mathrm{mean}(\mathrm{abs}(C_{\mathrm{all-corr}}))}, \tag{2}$$

$$\mathrm{PSR} = \frac{C_{\mathrm{peak}}}{C_{\mathrm{subpeak}}} \tag{3}$$

In the above equation, $C_{\mathrm{peak}}$ is the peak of the cross-correlation, $C_{\mathrm{all-corr}}$ is the cross-correlation matrix, and $C_{\mathrm{subpeak}}$ is the second peak of the cross-correlation. Figure 6c shows the result obtained using a neighborhood filter and co-filtering with the cross-correlation value and its derived parameter filter. Compared to Fig. 6b, which solely employs the neighborhood filter, incorporating the cross-correlation and derived parameter filter effectively eliminates erroneous outcomes within open-water regions.

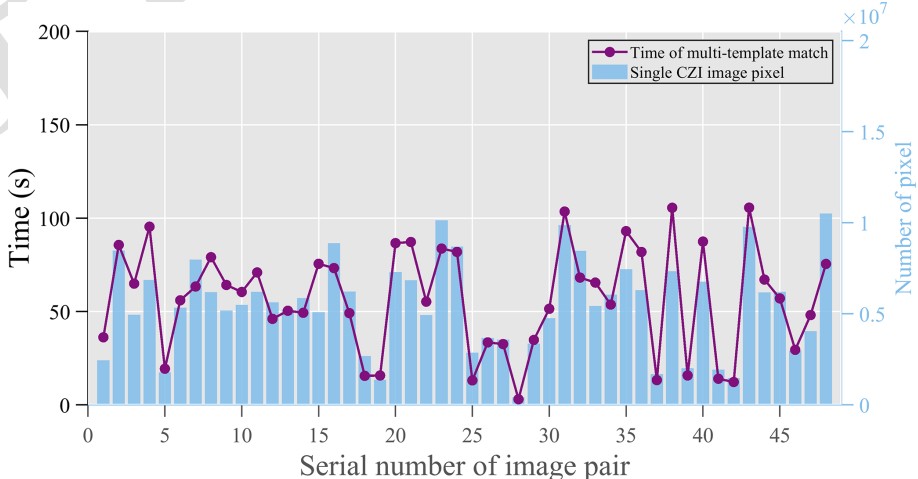

**Figure 4.** An example of a CZI image (average of bands 2, 3, and 4) before and after data enhancement and its grayscale histogram. Panel **(a)** is the original image and the color of its histogram is light blue. Panel **(b)** is the image after logarithmic stretching, and the color of its histogram is green. Panel **(c)** is the image after logarithmic stretching and CLAHE, and the color of its histogram is brown. Panel **(d)** is the grayscale histogram of the above images. The date of the image is from 7 April 2021 at 07:16:35 LT.

**Figure 5.** Relationship between the multi-template matching time and image size.

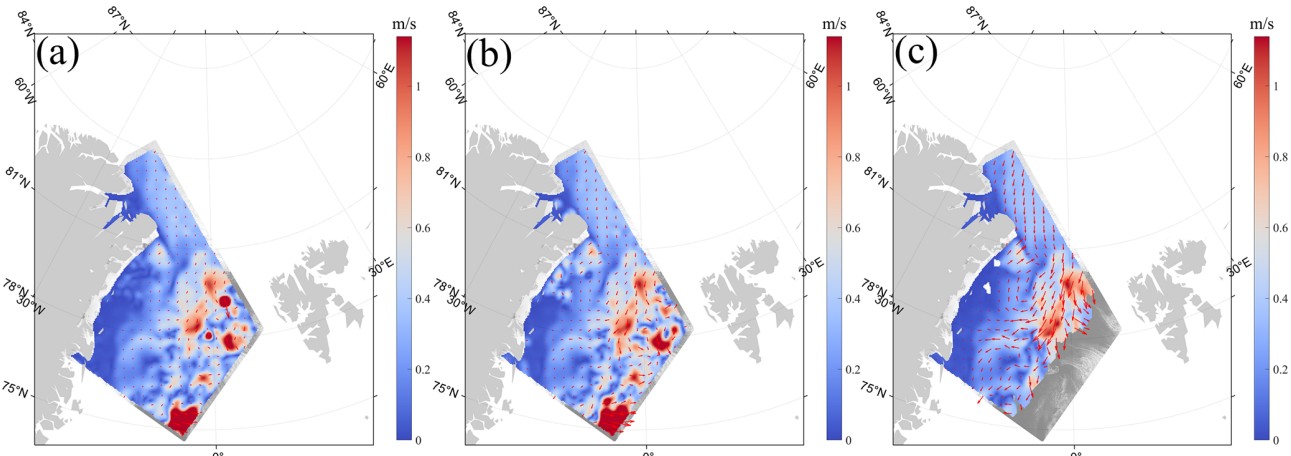

**Figure 6.** An example of quality control applied to the retrieved SID. (**a**) SID with no quality control. (**b**) SID after neighborhood filter (only). (**c**) SID after using neighborhood filter and cross-correlation value and its derived parameter filter.

### 3.3.2 Neighborhood filter

The neighborhood filter is commonly employed for postprocessing retrieved SID data by removing abrupt values indicative of incidental anomalous drifts that deviate from continuous sea ice movement patterns (Hyun and Kim, 2017; Girard-Ardhuin and Ezraty, 2012). In our study, two window sizes are designed for the neighborhood filter. Figure 6b shows the results of using the neighborhood filter only. Compared to Fig. 6a, the neighborhood filtering removes the regions (upper part of the image) with significant velocity differences in the SID, but only using the neighborhood filter is not incomplete. Combining the cross-correlation value and its derived parameter filter is necessary to comprehensively control the quality of the result.

### 3.4 Fill and smooth

The utilization of the correlation coefficient and its derived parameter filter and neighborhood filter introduces missing values to the results. Given that the sea ice is spatially distributed continuously, adjacent values are employed to fill in these gaps and smooth the outcomes during the final step of retrieval.

## 4 Results and validation

In this study, SID in the FS is retrieved from the CZI images with a resolution of approximately 4 km. The results are categorized into day level and hour level datasets, based on the time interval of the images and compared with the IABP buoy and the CMEMS SID product.

### 4.1 Comparison of results with CMEMS products

#### 4.1.1 SID with day level time intervals

Higher-resolution SID is retrieved using CZI images. The SID products based on radiometer or scatterometer values are incomparable with the retrieved SID due to the spatial resolution differences. Therefore, the SAR-based SID product (10 km) published by CMEMS is selected for comparison. The reliability of the CMEMS product has been verified in the released user manual (European Union–Copernicus Marine Service, 2015).

Figure 7 shows the $U - V$ difference between the retrieved SID (day level) and the CMEMS SID products, totaling 37 902 matched points. The $U - V$ differences between the retrieved SID and the product are relatively slight. The majority of the $U - V$ differences are less than $0.05\,\mathrm{m\,s^{-1}}$ and the majority of the scatter points are distributed in the center of the plot. The RMSE of $U$ and $V$ are 0.003 and $0.004\,\mathrm{m\,s^{-1}}$, respectively. A strong correlation between the product and the retrieved SID is shown by the histograms and accompanying fitted curves, with a normal distribution on the $x$ and $y$ axes.

Figure 8a shows the velocity difference between the retrieved SID and the CMEMS SID product, involving a total of 37 902 matched points. The retrieved SID demonstrates a strong correlation with the product in terms of velocity, as indicated by a high correlation coefficient of 0.913 and a fitted line that closely aligns with the ideal line. The statistics of the difference between the retrieved SID and the product are presented in Table 2, revealing RMSE of $0.003\,\mathrm{m\,s^{-1}}$ for the $U$ component, $0.004\,\mathrm{m\,s^{-1}}$ for the $V$ component, and $0.005\,\mathrm{m\,s^{-1}}$ for the velocity. These slight differences reflect the consistency between the retrieved SID and the product.

The size of the template and search area is associated with the spatial resolution of the result. To retrieve high-spatial-

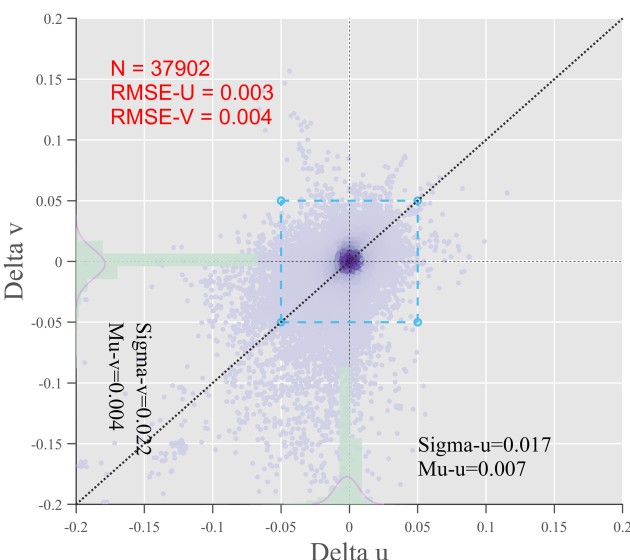

**Figure 7.** The $U - V$ difference between the retrieved SID (day level) and the product. The histograms on the $x$ and $y$ axes show the distribution of differences in the $U$ and $V$ directions, and the pink curves fit the histograms. The dashed blue box is $0.05\,\mathrm{m\,s^{-1}}$, and the color shading reflects the density of the scatter distribution.

resolution SID in the FS, a limited template size and search area is set in our study. Therefore, the maximum velocity of day level result is lightly smaller than the product. The effect of template size and search area on retrieving SID will be explored in future research.

Figure 8b shows the flow direction difference between the retrieved SID and the product, using a total number of 28 419 matched points. The correlation of the flow direction between the retrieved SID and the product is great, as indicated by a correlation coefficient of 0.897. The statistics of the difference in flow direction between the retrieved SID and the product are shown in Table 2.

Regarding the flow direction, the flow direction of sea ice motion in the FS predominantly ranges from $\pi/2$ to $3\pi/2$ (with the positive $y$ axis as the starting point in polar stereographic projection). The most abundant flow direction is $\pi$, which indicates the southern part of the FS. In the comparison of flow direction, it is observed that the shelf ice on the eastern coast of Greenland has a slower velocity (less than $500\,\mathrm{m\,d^{-1}}$) than the drift ice in the strait. Due to the potential inaccuracy associated with short sea ice displacement, a shelf ice mask was been established to exclude the slow-moving shelf ice. The impact of sea ice displacement on retrieval accuracy is analyzed in Sect. 5.

### 4.1.2  SID with the hour level time interval

Figure 9 illustrates the difference in $U - V$ between the retrieved SID (hour level) and the CMEMS SID product, with a total of 20 418 matched points. The scatter points distri-

bution of the $U - V$ differences in Fig. 9 appears to be significantly more dispersed than that in Fig. 7, suggesting a notable disparity between the retrieved SID and the product. The apparent difference may be associated with the spatial and temporal variability in the sea ice motion and will be explicitly analyzed in the discussion section.

### 4.2  Comparison of results with IABP buoys

### 4.2.1  SID with the day level time interval

The GPS position of the buoys serves as a crucial reference for validating the accuracy of the retrieved SID. Figure 10 illustrates the difference in velocity (Fig. 10a) and flow direction (Fig. 10b) between the result (day level) and buoys, with a total of 132 matched points. The correlation coefficient between the retrieved SID and the buoy of velocity is approximately 0.829, while it is approximately 0.778 for the flow direction. With the time interval of images being approximately 24 h (day level), there is a strong correlation between the retrieved SID and the buoy.

Table 3 shows the statistics of the retrieved SID and the buoy, which indicate that the bias is approximately $-0.005\,\mathrm{m\,s^{-1}}$, and the RMSE is approximately $0.031\,\mathrm{m\,s^{-1}}$ for velocity; moreover, for the flow direction, the bias is $0.002\,\mathrm{rad}$, and the RMSE is $0.009\,\mathrm{rad}$.

### 4.2.2  SID with the hour level time interval

Figure 11 shows the difference in velocity (Fig. 11a) and flow direction (Fig. 11b) between the result (hour level) and buoys, with a total of 74 matched points. The correlation coefficient of the velocity between the retrieved SID and the buoy is 0.926, and the correlation coefficient of the flow direction between the retrieved SID and the buoy is 0.889. When the time interval of the images is less than 6 h (hour level), there is a satisfactory correlation between the retrieved SID and the buoy.

Table 4 provides the statistics of the retrieved SID and the buoy, revealing negligible bias values ($\sim 0\,\mathrm{m\,s^{-1}}$) along with a RMSE of $0.036\,\mathrm{m\,s^{-1}}$ for velocity. The bias is $0.003\,\mathrm{rad}$, and the RMSE is $0.010\,\mathrm{rad}$ for flow direction. Wang et al. (2022) evaluated six SID products over eastern Greenland with buoys and found that the mean absolute errors (MAEs) of velocity for these products are ranged from $0.016$–$0.120\,\mathrm{m\,s^{-1}}$, and the MAEs of flow direction ranged from $0.300$–$0.973\,\mathrm{rad}$. Combined with Tables 3 and 4, it can be seen that the velocity and flow direction MAEs of our result are smaller than SID products based on the radiometer, scatterometer, and AVHRR.

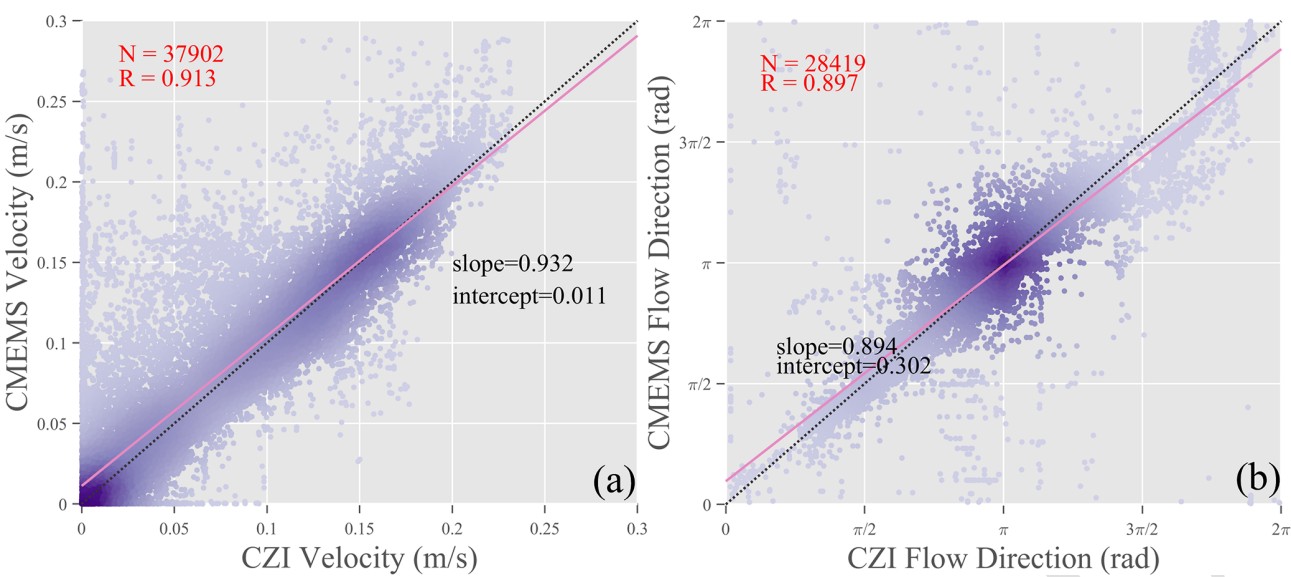

**Figure 8.** The difference in velocity **(a)** and flow direction **(b)** between the retrieved SID (day level) and the product. The dashed line denotes the ideal line, and the pink line denotes the fit line.

**Table 2.** Validation of the retrieved SID (day level) with the product.

| Delta | No. of match CE3 points | Bias | MAE | SD | RMSE |
|---|---|---|---|---|---|
| $U$ (m s$^{-1}$) | | $-0.003$ | 0.009 | 0.017 | 0.003 |
| $V$ (m s$^{-1}$) | 37 902 | $-0.004$ | 0.013 | 0.024 | 0.004 |
| Velocity (m s$^{-1}$) | | $-0.005$ | 0.014 | 0.027 | 0.005 |
| Flow direction (rad) | 28 419 | 0.009 | 0.147 | 0.369 | 0.009 |

## 5 Discussion

### 5.1 Analysis of the differences between the retrieved SID and the CMEMS SID product

#### 5.1.1 Comparison of individual retrieved SID and the product

The statistics of the retrieved SID and the CMEMS SID product are analyzed in Sect. 4. In this section, we compare the retrieved SID with the corresponding product in chronological order. Figure 12a shows the velocity correlation coefficient curve between the day level results and the product. Most of the correlation coefficients are greater than 0.7, and the bars of the 95 % confidence level are idealistic. When the correlation coefficient is high, the bar of the 95 % confidence level is also behave relatively small. However, the 11th and 12th data pairs exhibited significantly lower correlation coefficients, primarily due to the involvement of fewer matched points (fewer than 500). Additionally, the correlation coefficient of the second data pair is much lower than that of other data pairs, due to the dense cloud cover in the CZI images, which affects the retrieval of SID. Figure 12b shows the correla-

tion coefficient curve of the flow direction between the day level results and the product. The correlation coefficient of the 10th pair is negative and considerably different from that of other data because there are fewer matched points (fewer than 500). Furthermore, the retrieved SID of the 10th pair is located on the northeastern coast of Greenland. As CE4 the region where TPD encounters the Greenland coastline, the sea ice motion in the region is mutable. The sea ice motion trend in the area is highly variable due to the influence of wind and other forces. The finding reveals that interactions between sea ice in the northern FS, where MYI congregates, can drastically affect the flow direction of the SID. Figure 12c shows the velocity correlation coefficient curve between the hour level results and the product. The significantly lower correlation coefficient of the first pair of data can be attributed to the fact that there are not enough matched points (less than 200), and the confidence level bar of the first pair is also unsatisfactory. Figure 12d shows the correlation coefficient curve of the flow direction between the results (hour level) and the product. The correlation between the retrieved SID (hour level) and the product in the flow direction is worse when comparing Fig. 12b and d.

**Table 3.** Validation of the retrieved SID (day level) with IABP buoys.

| Delta | No. of match points | Bias (relative deviation$_{mean}$) | MAE | SD | RMSE |
|---|---|---|---|---|---|
| Velocity (m s$^{-1}$) | 132 | −0.005 (−1.330 %) | 0.018 | 0.031 | 0.031 |
| Flow direction (rad) | | 0.002 (0.149 %) | 0.003 | 0.009 | 0.009 |

**Table 4.** Validation of the retrieved SID (hour level) with IABP buoys.

| Delta | No. of match points | Bias (relative deviation$_{mean}$) | MAE | SD | RMSE |
|---|---|---|---|---|---|
| Velocity (m s$^{-1}$) | 74 | 0.000 (1.998 %) | 0.021 | 0.036 | 0.036 |
| Flow direction (rad) | | 0.003 (0.154 %) | 0.006 | 0.010 | 0.010 |

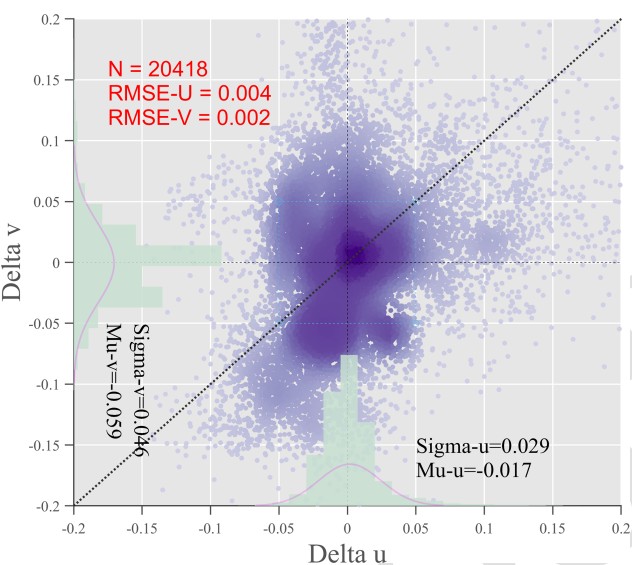

**Figure 9.** The $U - V$ difference between the retrieved SID (hour level) and the product. The histograms on the $x$ and $y$ axes show the distribution of differences in the $U$ and $V$ directions, and the pink curves are a fit to the histograms. The dashed blue box is 0.05 m s$^{-1}$, and the color shading reflects the density of the scatter distribution.

The bias and standard deviation of the velocity between the day level results and the CMEMS SID product are shown in Fig. 13a. The velocity bias of the two SID datasets is less than $\pm 0.05$ m s$^{-1}$, demonstrating a high degree of consistency. When the time interval is large (12th and 28th pairs) or there are fewer matched points (3rd, 10th, 11th, and 13th pairs), the bias and standard deviation between our results and the products increase. Figure 13b shows the bias and standard deviation of the flow direction between the day level results and the product. The bias of the flow direction is slight. When there are fewer matched points (3rd, 10th, and 23rd pairs), the bias and standard deviation of the flow direction become larger. Figure 13c shows the velocity bias and standard deviation between the hour level results and the product. Although most of the biases are smaller than

0.15 m s$^{-1}$, the velocity bias is significantly more substantial than that of the day level result. The bias and standard deviation of the flow direction between the results (hour level) and the product are shown in Fig. 13d. By comparing Fig. 13a and b with Fig. 13c and d, the day level results are much less divergent from the product than the hour level results.

### 5.1.2 Analysis of the difference

The day level results are compared to the CMEMS SID product in Figs. 7 and 8, revealing the slight difference in $U$, $V$, velocity, and flow direction. The statistics in Table 2 also illustrate the slight difference between our results and the product. However, Fig. 12 shows a significant difference between the hour level result and the product. The retrieval methods used in our study differed slightly from the method of the product, but both algorithms are based on the MCC. This discrepancy may be attributed to either the time interval of SAR images used in the product or the time interval between the two kinds of SID.

Figure 14 shows the time interval of SAR images used by the CMEMS SID product. The blue histogram represents the time interval of the SAR images used by the product, which is compared to the day level result, and the purple histogram represents the time interval of the SAR images used by the product, which is compared to the hour level result. Figure 14 shows that the blue and purple histogram distributions are nearly identical. Most time intervals of SAR image used by the product are less than 6 h, and only a few are approximately 20 h. The consistent distribution trend of the two kinds of histograms indicates that the time intervals of the SAR images used by the product are basically the same. The distribution of the histograms explains that the difference between the retrieved SID and the product is not relevant to the time interval of the SAR images used by the product.

We attribute the difference between the hour level result and the CMEMS SID product to the temporal and spatial variability in the sea ice motion. Figure 15a and c show the variations in the velocity and flow direction of 17 different buoys with different time intervals. The date of the

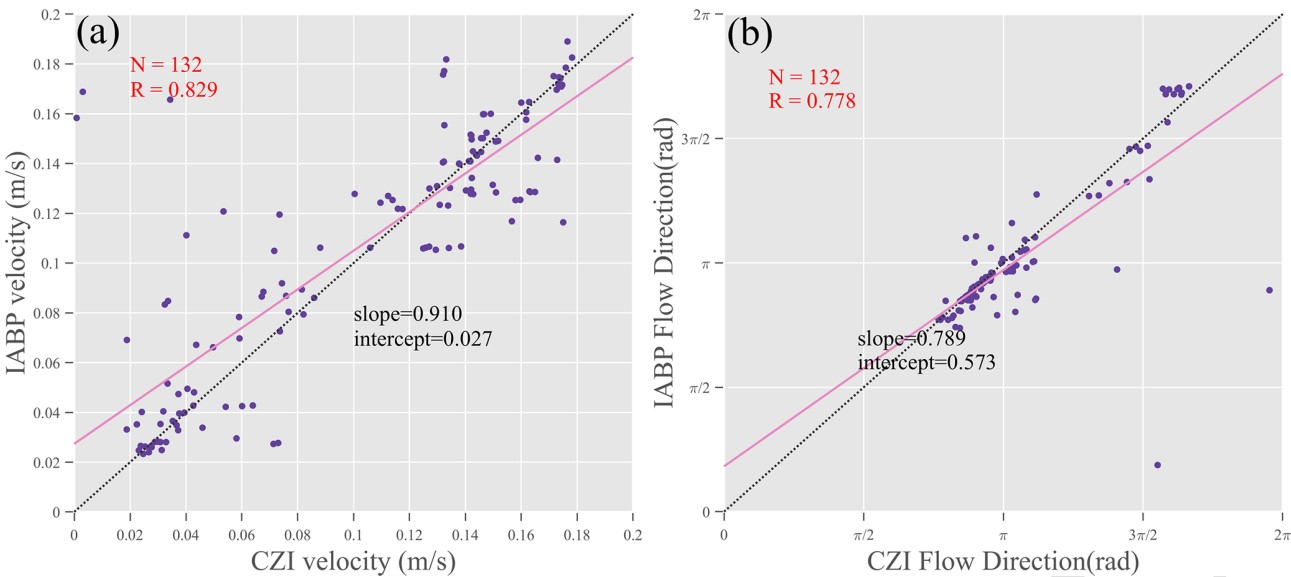

**Figure 10.** Comparisons between the retrieved SID (day level) and IABP buoys in velocity **(a)** and flow direction **(b)**. The dashed line denotes the ideal line, and the pink line denotes the fit line.

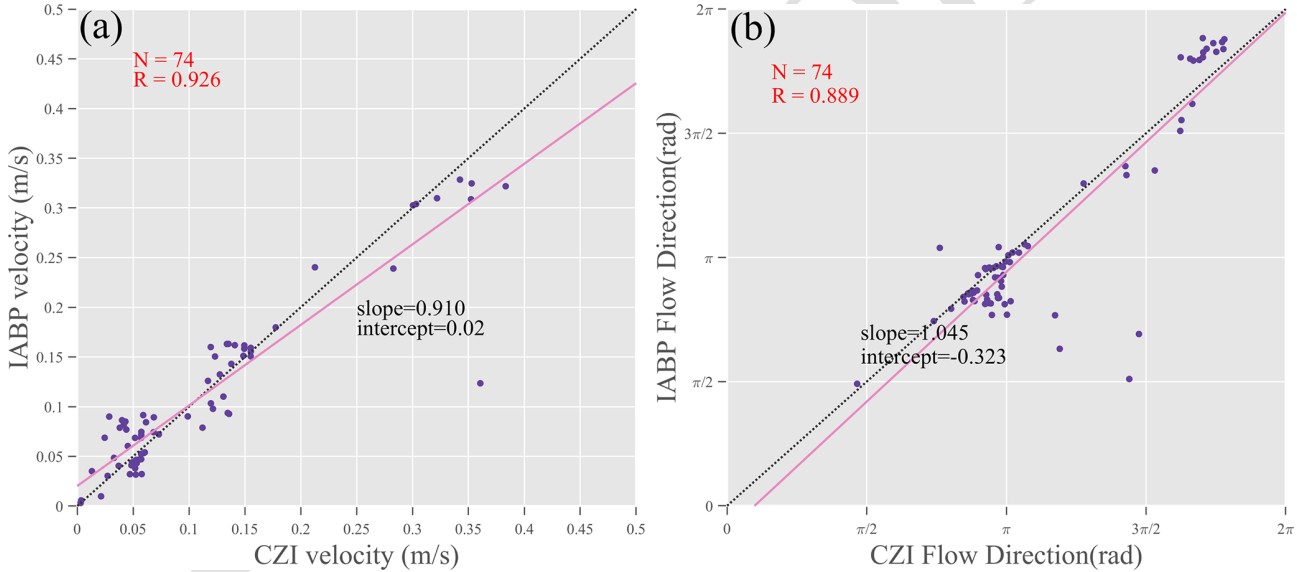

**Figure 11.** Comparisons between the retrieved SID (hour level) and IABP buoys in velocity **(a)** and flow direction **(b)**. The dashed line denotes the ideal line, and the pink line denotes the fit line.

day level CZI images is from 6 April 2021 at 06:10:12 LT and 7 April 2021 at 07:16:35 LT. The first date of the hour level CZI images is from 6 April 2021 at 06:10:12 LT and 6 April 2021 at 09:29:35 LT; the second date of the hour level CZI images is from 6 April 2021 at 06:10:12 LT and 6 April 2021 at 11:10:03 LT; and the third date of the hour level CZI images are from 6 April 2021 at 09:29:35 LT and 6 April 2021 at 11:10:03 LT. The triangles in Fig. 15a represent the velocity of the buoys, and there is a significant difference in velocity between the day level buoys and the hour level buoys. The difference in the velocity between hour level buoys is less than the difference between hour level buoys and day level buoys, but the difference still exists. The same trend is also found in the difference in the flow direction, as shown in Fig. 15c. Figure 15b and d show the differences in velocity and flow direction between our results and the buoys, and the day level result deviates less from the buoy. Additionally, different buoys show various differences from our results, which are related to the location of the buoys.

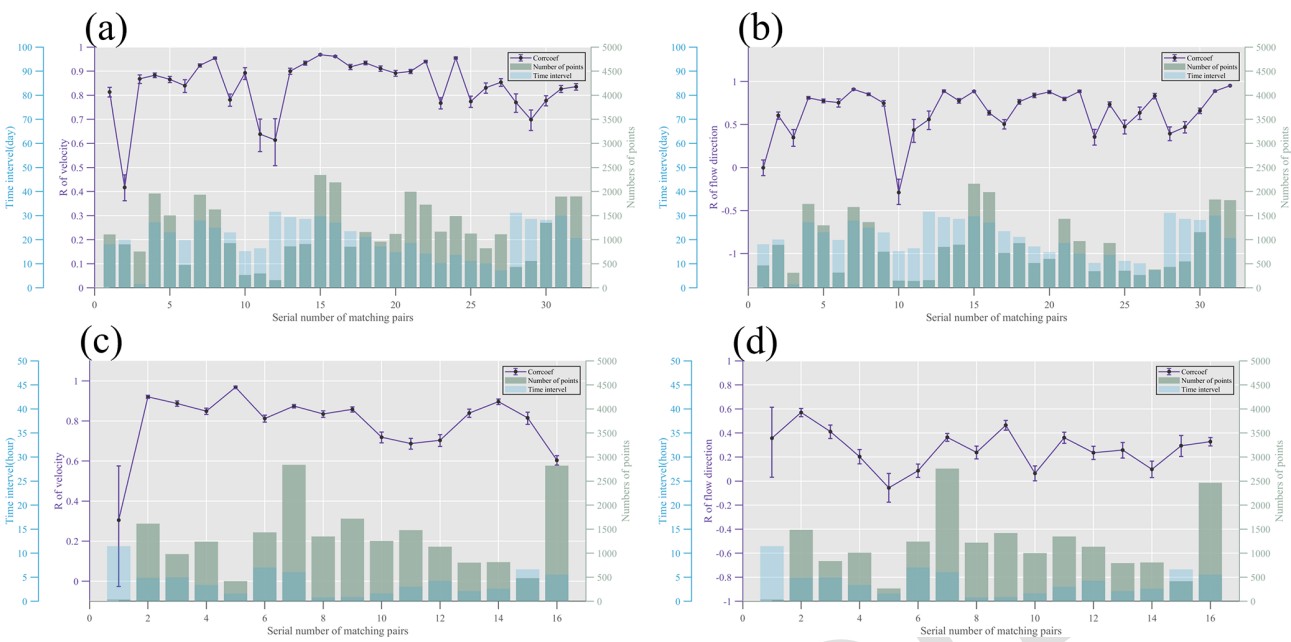

**Figure 12.** The correlation coefficient and 95 % confidence level bar of correlation coefficient between the retrieved SID and the CMEMS SID product. **(a, b)** The correlation coefficient between the retrieved SID (day level) and the product in terms of velocity and flow direction. **(c, d)** The correlation coefficient between retrieved SID (hour level) and the product in terms of velocity and flow direction. The green histogram (right axis) shows the number of matched points between the retrieved SID and the product, and the blue histogram (the first left axis) shows the time interval between the retrieved SID and the product.

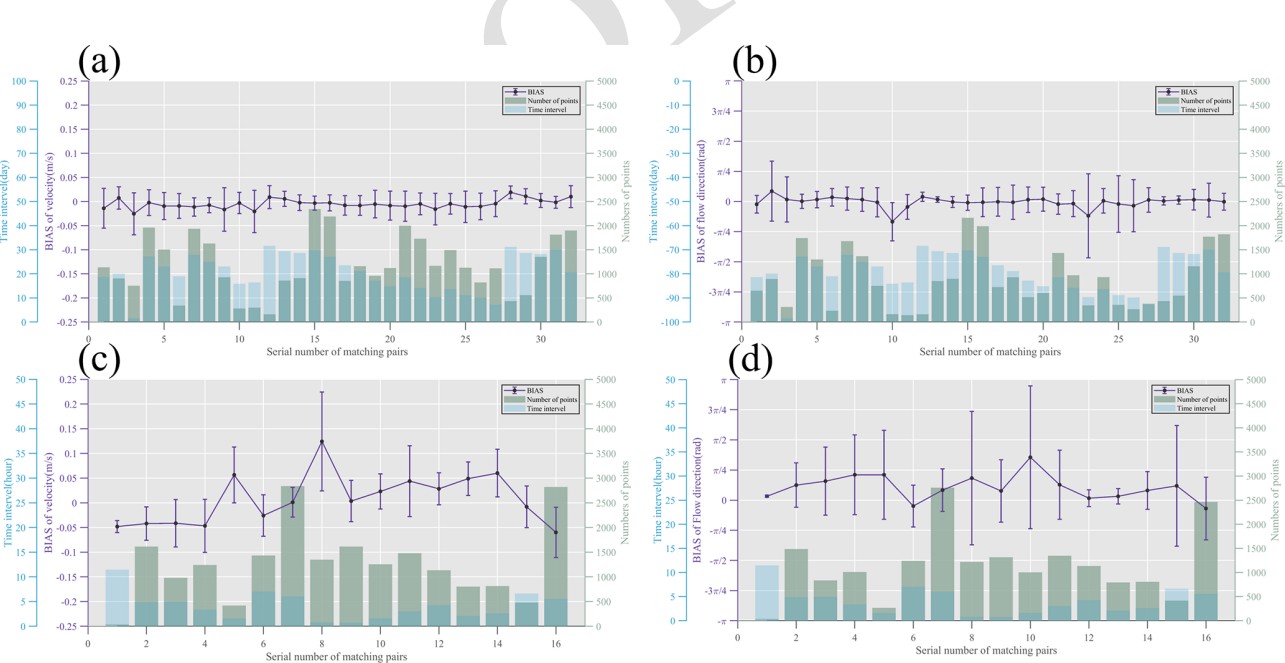

**Figure 13.** The bias and standard deviation between the retrieved SID and the CMEMS SID product. **(a, b)** The bias between the retrieved SID (day level) and the product in terms of velocity and flow direction. **(c, d)** The bias between retrieved SID (hour level) and the product in terms of the velocity and flow direction. The green histogram (right axis) shows the number of retrieved SID pixels and the product matched points, and the blue histogram (the first left axis) shows the time interval between the retrieved SID and the product.

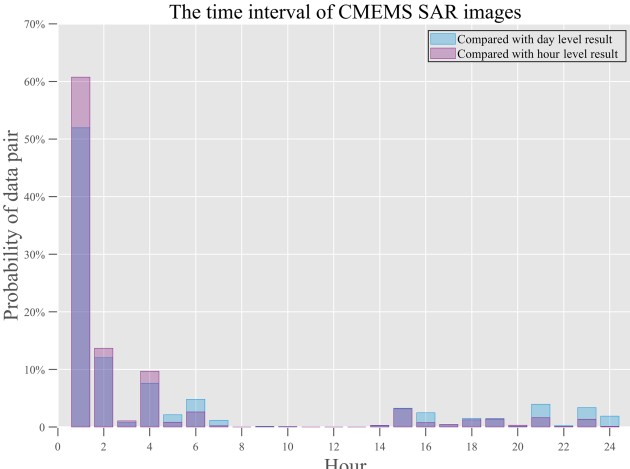

**Figure 14.** The time interval of SAR images used by the CMEMS SID product during the comparison with two different results.

The velocity and flow direction differences in the buoys with different time intervals suggests that SID has large spatial and temporal variability. There is a distinct variability in velocity and flow direction when the SID is retrieved us-
5 ing images with short time interval. When the drift vectors retrieved with longer time interval are compared with those vectors retrieved at shorter intervals, differences are noted. This phenomenon is a consequence of the spatial and temporal variability in the SID. When comparing the drift vec-
10 tors retrieved with two different short time intervals, in extreme situations, a greater difference will be presented. The variability in the sea ice motion is more complex and unpredictable on a small timescales. The complex spatial and temporal variability in the SID explained the discrepancy be-
15 tween our results and the CMEMS SID product.

Despite the spatial and temporal variability in the sea ice motion, it is still important to validate our retrieved results using the product. Furthermore, as long as a sufficiently large time interval is employed, the impact of SID variability di-
20 minishes, which results in consistent drift trend being observed. Utilizing SID products for evaluating the retrieved result remains valid, while providing valuable references through consistency and difference ratings.

## 5.2 Effect of different time intervals on retrieving SID

In Sect. 5.1, the consistency and discrepancy between the result and the CMEMS SID product is discussed, revealing the spatial and temporal variability in the sea ice motion. In this section, we explore the impact of different time intervals on retrieving SID by analyzing the validation of the buoys and
the statistics of the quality control parameters. The time interval of CZI images is a factor that affects the accuracy of retrieving SID. On the one hand, elaborate sea ice motion can be tracked using images with short time interval; on the other hand, retrieving SID from images with longer time interval

**Table 5.** Statistics of quality control parameters at different time intervals.

| Parameter | Mean | SD | Type |
|-----------|--------|-------|------------|
| $R$ | 0.372 | 0.199 | Day level |
|  | 0.473 | 0.225 | Hour level |
| PMR | 9.338 | 4.857 | Day level |
|  | 13.262 | 6.198 | Hour level |
| PSR | 1.620 | 0.838 | Day level |
|  | 2.016 | 0.980 | Hour level |

provides more stable result. Therefore, it is crucial to discuss 35 the appropriate time interval for retrieving SID in the FS.

Section 3.3 describes the three parameters used for quality control: $R$, PMR, and PSR. Table 5 presents the statistics of these parameters with different time intervals. Each quality control parameter's mean value of the hour level result 40 is greater than that of the day level result, which indicates that the hour level result has the higher quality. However, the standard deviation of the quality control parameters exhibit opposite trend. The standard deviation of the quality control parameters for the day level result is smaller than for the hour 45 level result. The statistics indicate that the day level result remains relatively stable, despite having slightly poorer quality.

In the validation with the buoy, the correlation of the day level result is slightly lower than that of the hour level result in velocity. In terms of flow direction, the correlation of 50 the day level result is also lower than that of the hour level result. The retrieval of day level SID is confronted with the effects of large drift distances, wind change, light variation, and cloud cover, which are disadvantageous for retrieving. The statistics in Tables 3 and 4 also show that the hour level 55 result possesses better accuracy than the day level result. Images with short time interval provide a more stable scenes for retrieving SID, thus the hour level result possesses finer accuracy.

In our study, the dataset was divided into two groups ac- 60 cording to the image time interval. In comparison with the buoy, both results achieved great accuracy. According to the analysis of the day level result and the hour level result, we believe that retrieving SID using images with a short time interval will provide SID with better accuracy. Our proposi- 65 tional time interval is shorter than the time interval proposed by Qiu and Li (2022), who suggested using images with 20 h time interval to retrieve SID in the FS from January to June. We attribute these divergent conclusions on suitable time interval due to the limitations of our dataset. In future studies, 70 we are planning to incorporate HY1-C data and investigate the impact of varying time intervals on SID retrieval.

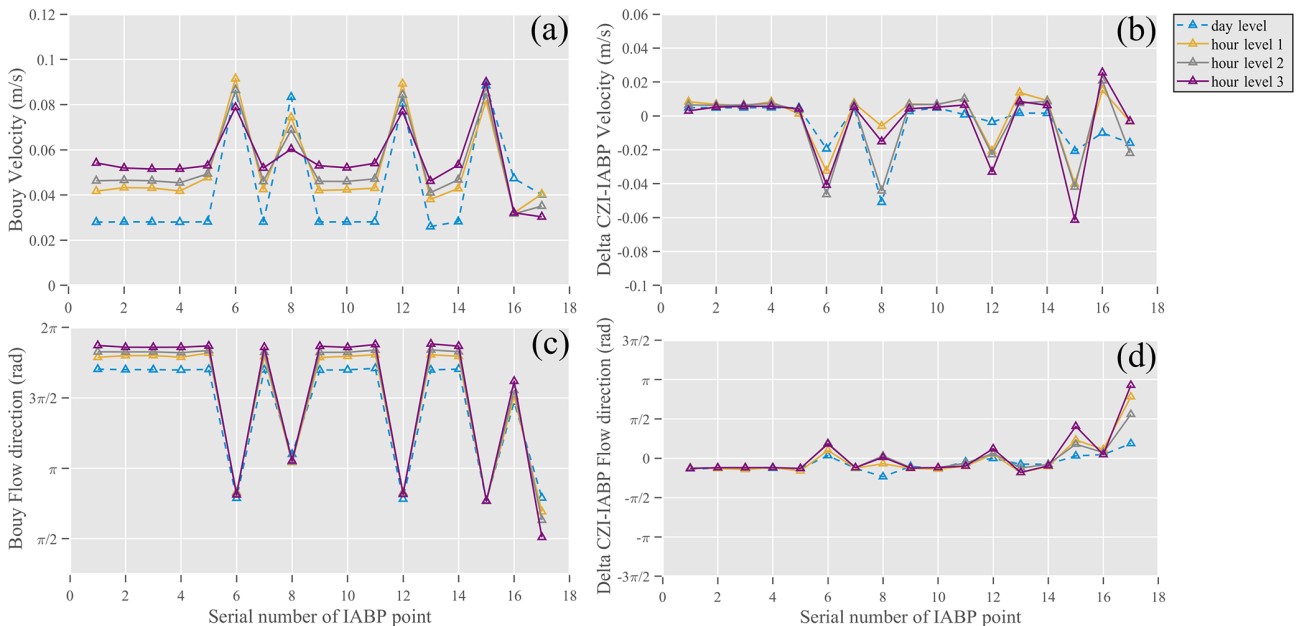

**Figure 15.** The buoy velocity **(a)** and flow direction **(c)** and the difference (velocity **(b)**; flow direction **(d)**) between the retrieved SID and buoys. The lines in blue represent the SID and buoy vectors with a day level time interval, and the lines in yellow, gray, and purple represent SID and buoy vectors with an hour level time interval on the same day. The *x* axis shows the different matched SID and buoys. The *y* axis in panels **(b)** and **(d)** shows the difference between SID and buoys.

### 5.3 The potential of retrieving sea ice drift from image with cloud cover

Retrieving SID using optical imagery can be affected by factors such as cloud cover, lighting, and other variables. Sobel edge detection is used to enhance the texture of sea ice. We found that SID result is also achieved for sea ice under cloud cover after using Sobel edge detection. Figure 16 shows two examples of SID retrieved from images with cloud cover. The first example uses hour level time interval images for retrieval, with Fig. 16a taken on 27 April 2021 at 09:27:33 LT and Fig. 16b taken on 27 April 2021 at 11:07:20 LT. The second example uses day level time interval images for retrieval, with Fig. 16c taken on 2 May 2021 at 03:20:35 LT and Fig. 16d taken on 3 May 2021 at 06:07:37 LT. In the second example, Fig. 16 and e are also covered by thick cloud cover in addition to thin cloud cover, which more severely obscures the sea ice texture.

Figure 16a and b show the cloud cover and open water by visual interpretation. Figure 16c shows the SID vectors derived from the images, where the vectors with Sobel edge detection in the preprocessing are in red, and the vectors without Sobel edge detection are in blue. A comparison of these two types of vectors reveals a more consistent trend of SID with a homogeneous velocity distribution achieved using Sobel edge detection, which demonstrates that edge detection promotes the retrieval algorithm and illustrates the potential for optical imagery to retrieve SID using images with thin cloud cover. Figure 16d, e, and f show another pair of results

using CZI images with the day level time interval. In contrast to the first pair of results, the CZI images (Fig. 16d, e) have more cloud cover, and the variation in the cloud distribution is quite apparent. Comparing the vectors with two different colors, the red vector is better, even providing results in parts of the area under thick cloud cover. Thin cloud cover is distributed in lower-right corner of the image, but the algorithm does not give a credible result due to the strongly fragmented sea ice. Analytically, edge detection strengthens the ability of algorithm to retrieve SID from images with cloud. In addition, Fig. 16 similarly shows the enhancement of the separating capacity for open water in the algorithm using detection. The red vectors in Fig. 16c and f are mainly distributed within the sea ice extent.

To quantify the potential of retrieving SID from image with cloud cover, we use the whiteness index (Gómez-Chova et al., 2008) and multiscale edge-preserving decompositions (MEDs) (Farbman et al., 2008) to detect the cloud pixels (Kang et al., 2018). Figure 17a and b show the number of no-cloud pixels and the number of retrieved SID pixels. For the day level results, in addition to the influence of cloud pixels, dramatic shape changes in the sea ice also hinder the retrieval of SID. As shown in Fig. 17a, only a few (1st, 15th, 21st, 30th, 31st, and 32nd) of the retrieved SID pixels are beyond the no-cloud pixels. For the 32nd image pair, the number of SID pixels is 120 % greater than the number of no-cloud pixels, possibly because the cloud in the images is quite thin. For the hour level results, images with a short time interval provide a stable scene for retrieving. Thus, many of the retrieved

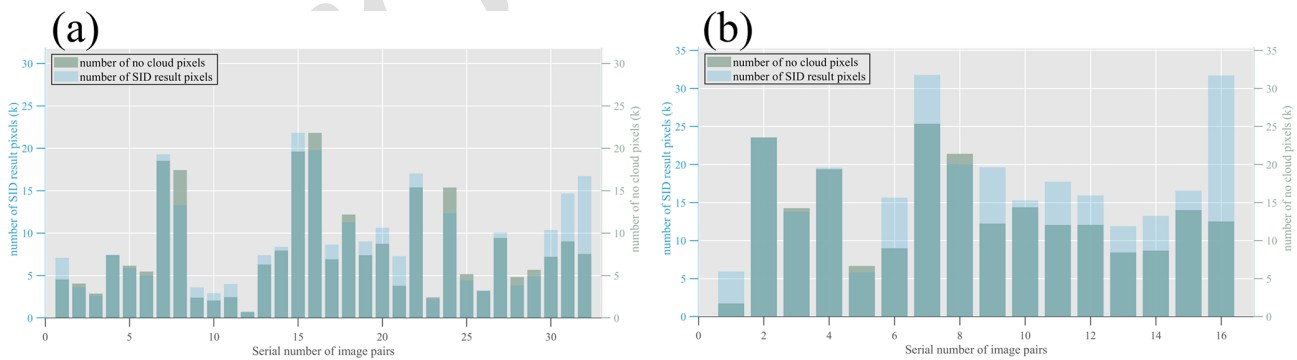

**Figure 16.** Two examples of retrieving sea ice drift from image with cloud cover. Panels **(a)**, **(b)**, and **(c)** are a pair of results, and panels **(d)**, **(e)**, and **(f)** are another pair of results. The arrows in panels **(c)** and **(f)** are diluted for direct viewing.

**Figure 17.** The number of no-cloud pixels (green) and the retrieved SID pixels (blue) (**a**, day level result; **b**, hour level result).

SID pixels (1st, 6th, 7th, 9th, and 11th–16th) are greater than the no-cloud pixels, as shown in Fig. 17b. For the 16th image pair, the number of retrieved SID pixels is 150 % greater than the number of no-cloud pixels; this is because the cloud is slight and the distributions of cloud in two images are similar. The average of the SID pixels beyond the no-cloud pixels is 10.109 % and 28.920 % for the day level result and the hour level result, respectively. In conclusion, our approach has the potential to retrieve SID effectively with the images containing cloud pixels. The result shows that with less strict restriction and appropriate preprocessing, images with cloud cover could be used to obtain credible and dense SID.

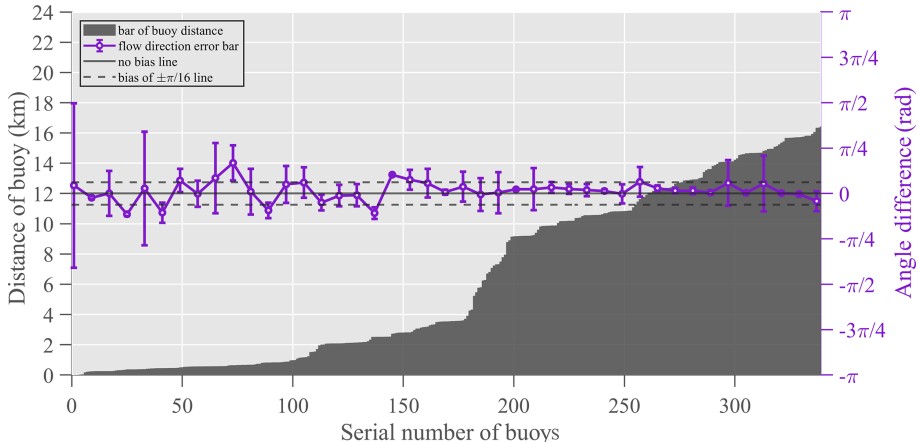

**Figure 18.** Relationship between the flow direction and the displacement of the buoys.

## 5.4 Factors affecting the accuracy of flow direction retrieval

Qiu and Li (2022) concluded that the flow direction accuracy of the retrieved SID is associated with the magnitude of the velocity. In our study, when dividing the dataset into hour level and day level by time interval for comparison with the buoys, an inconsistent conclusion was found when analyzing the accuracy of flow direction. Compared with the buoy data, the statistics in flow direction of the SID at day level is slightly smaller than the SID at hour level. However, since the coverage areas of both datasets are repetitive, the magnitudes of velocity retrieved from both datasets should be the same. Thus, it is not the velocity magnitude of the SID that affects the accuracy of the flow direction retrieval.

We consider that the distance of sea ice motion affects the accuracy of the flow direction retrieval. Due to stochastic error, the identification of the highest correlation point deviates to the optimal point. Consequently, the retrieved flow direction is constrained to the vicinity of the optimal point. The variability in the flow direction retrieval with a longer distance is more stable than with shorter distance. Retrieving the flow direction precisely with small displacement is challenging, and the time interval needs to be increased to achieve longer sea ice displacements, although this can also lead to other disadvantages (e.g., cloud and ice disintegration) for SID retrieval. Figure 18 shows the relationship between the drift distance of the buoys and the difference between the retrieved SID and the buoy in the flow direction. We generated a total of 206 matched points and merged every 5 points to calculate the standard deviation. As the buoy displacement increases, the bias and standard deviation of the flow direction decrease, which indicates that the accuracy of the retrieved flow direction is affected by the distance of sea ice motion. A larger displacement results in more precise flow direction retrieval but also requires longer time interval images which will introduce disadvantageous factors for the algorithm.

## 5.5 Factors affecting the accuracy of velocity retrieval

The study area spans the Arctic Basin to the Fram Strait, where the velocity of the SID generally presents complicated variations. The maximum magnitude of buoy-derived ice velocity can reach to $0.64\,\mathrm{m\,s^{-1}}$ in the FS (Lei et al., 2016). Sea ice in the north of the FS or close to the land has a relatively low velocity. It is imperative to make further research on the accuracy of retrieving SID with different velocity.

In our study, the data were from March to May 2021, and we generated a total of 206 matched points and merged every 4 points to calculate the standard deviation of the velocity difference. We further analyzed the relation between the accuracy of retrieved velocity and the buoy velocity. As shown in Fig. 19, the bar shows that the velocity of buoys that we used in validation ranged from 0 to $0.32\,\mathrm{m\,s^{-1}}$, and the line of bias and standard deviation shows the accuracy of the retrieved velocity, which is associated with the velocity of the buoys. As the velocity of the buoys increases to $0.1\,\mathrm{m\,s^{-1}}$, both the bias and standard deviation also increase. Although the bias of retrieved velocity occurs within in an ideal range as the velocity of the buoys increases, the standard deviation of the velocity difference becomes erratic. In general, sea ice motion with great velocity appears unstable in the images. Our algorithm still needs improvement in retrieving SID with high velocity.

Additionally, we delve into the correlation between the distance of sea ice motion and the accuracy of the retrieved velocity. A significant challenge to the accuracy of velocity retrieval, as emphasized by Lavergne et al. (2010), is the presence of quantification noise in template matching. Despite the acknowledged presence of the quantification noise, our result surprisingly does not reveal its discernible effects, and a lack of a statistically significant correlation between

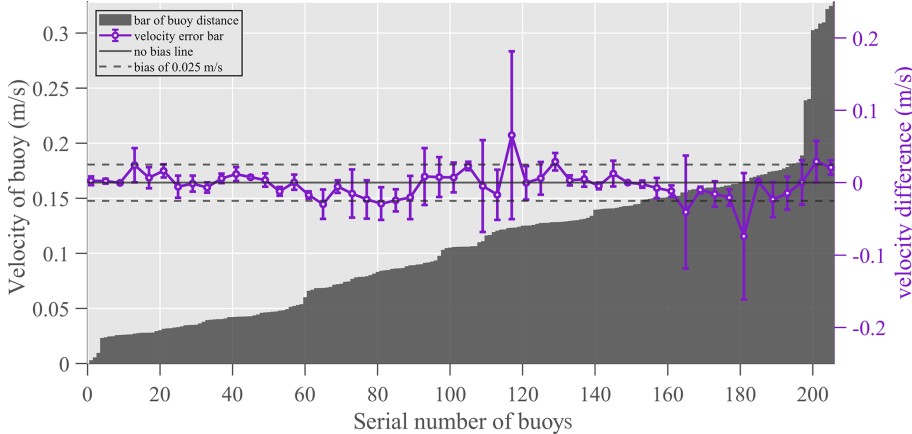

**Figure 19.** Relationship between the velocity difference and the velocity of the buoys.

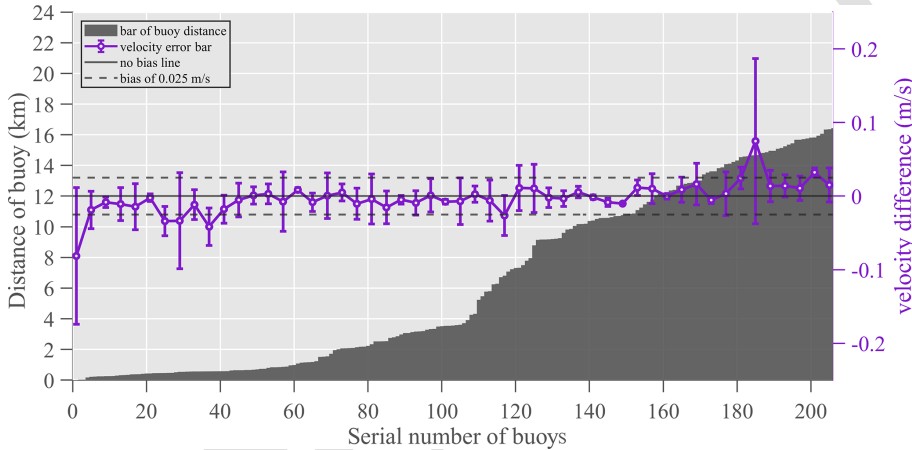

**Figure 20.** Relationship between the velocity difference and the displacement of the buoys.

the accuracy of the retrieved velocity and buoy distance is observed (as shown in Fig. 20).

Several explanations underpin this result. First, the spatial resolution of the resampled CZI imagery is 300 m, which is a notable improvement for the ice surface feature observation compared to the kilometer-level resolution of radiometer. The higher spatial resolution is posited to be a key factor limiting the manifestation of quantification noise. The topographic features of sea ice are often quantified by the surface roughness and form drag (Arya, 1973, 1975) and those observed surface features have impact on ice drift speed (Zu et al., 2021). The form drag can be decomposed into the contributions from ridges and snow dunes, and it is associated with the momentum exchange between sea ice and the lower atmosphere forced by wind (Zhang et al., 2024 TS7). The topographic features (including height and spacing) within the optical imagery need to be described quantitatively in the future work. Second, the strong sea ice motion in the FS renders sea ice dynamics distinctly visible in the image, indirectly suppressing the appearance of quantification noise. Fi-

nally, our application of subpixel estimation for the precise localization of maximum correlation values results in an exact determination of the maximum correlation location. This refinement contributes to improve the accuracy of retrieving SID. In conclusion, high spatial resolution data and subpixel estimation are able to suppress the negative effects caused by quantification noise.

## 5.6 The parameters for quality control

To validate the effectiveness of quality control parameters, a comprehensive examination was undertaken, involving a comparative analysis between the quality control parameters and buoy validation of the SID. The red line in Fig. 21 shows the $R$ of the validation points, and to identify the relationships between $R$ and the other parameters, the data are sorted in ascending order by $R$. The bar in orange and bar in blue represent the PMR and PSR, respectively. Figure 21 revealed a pronounced congruence, wherein the $R$, PMR, and PSR exhibited coherent consistency. A positive correlation was discerned, indicating that the increase in $R$ proportionally

accompanies the increase in the PMR and PSR. The magnitude of $R$ has been extensively used as a quality control parameter for SID retrieval (Qiu and Li, 2022; Haarpaintner, 2006; Ezraty et al., 2007). Given the observed synchronous trends of the PMR, PSR, and $R$, it is evident that the PMR and PSR collectively demonstrate efficacy in quality control. In Fig. 21, the error bar in purple shows the difference between the retrieved SID and buoys. The bias and standard deviation of the retrieved SID diminish as $R$, the PMR, and the PSR increase, which indicates the effectiveness of these parameters for revealing the reliability of the result.

The coherence between the quality control parameters and the validation of the buoys is illustrated in Fig. 21, providing a foundation for the meticulous examination of the SID quality. As one of the main outlets of Arctic sea ice in the Atlantic Ocean, the sea ice concentration in the FS is inferior to that in the Arctic Basin (Peng and Meier, 2018; Wang et al., 2020), and the sea ice in the south and north of the strait render different morphologies in the images.

To investigate the quality of SID over the strait, we selected 80° N as the segmentation line and calculated the quality control parameters of SID in north and south of the segmentation line. Figure 22 shows the stacking bars of different parameters in different regions. For the sake of visualization, we normalize the PMR and PSR. The average values of these quality control parameters indicate that SID to the north of 80° N have higher quality. A comparison of quality control parameters for results with different time intervals reveals that the hour level result possesses higher quality. Besides, the hour level result also has a greater mean value for quality control parameters to the south of 80° N, where the sea ice is dispersive. High-concentration sea ice constrains the variability in the sea ice motion and provides better spatial consistency, which is beneficial for retrieving SID. Similarly, the variability in the sea ice motion with short time interval is inconspicuous, providing a favorable scene for the algorithm. The sea ice kinematics in marginal ice zone are intricate, and the effects of wave on the fragmentary sea ice motion are evident (Williams et al., 2013, p. 1), which increases the uncertainty in the retrieval. The utilization of short time interval images for SID retrieval proves instrumental in enhancing the quality of monitoring sea ice motion within marginal ice zones.

## 6 Conclusion

The retrieval of SID in the FS is complicated due to the geographic specificity. Sea ice drift within the Arctic Basin is dominated by two trends, the BG and TPD. The FS is the outlet of TPD, as well as the intersection of the Atlantic and Arctic. The heavily fragmented sea ice in the strait, coupled with its fast velocity, significantly impacts the retrieval of SID. Our method, utilizing the multi-template matching and subpixel estimation approach for retrieving SID in the

FS, demonstrates great accuracy. We use 111 scenes of CZI images and obtain 48 sets of image pairs for analysis. The dataset is categorized into hour level and day level, according to the time interval. We found that the day level result are consistent with the CMEMS SID product; the hour level result is slightly different from the CMEMS SID product, due to the spatial and temporal variability in the sea ice motion.

Validation is also conducted using IABP buoys. For the velocity, the bias of the day level result and hour level result with the buoys is $-0.005$ and $0.000\,\mathrm{m\,s^{-1}}$, respectively, and the RMSE of the day level result and hour level result with the buoys is $0.031$ and $0.036\,\mathrm{m\,s^{-1}}$, respectively. For the flow direction, the bias of the day level result and hour level result with the buoys is $0.002$ and $0.003\,\mathrm{rad}$, respectively, and the RMSE of the day level result and hour level result with the buoys is $0.009\,\mathrm{rad}$ and $0.010\,\mathrm{rad}$, respectively. The accuracy of the retrieved SID surpasses that of the SID products retrieved from AVHRR.

The method with multi-template matching and subpixel estimation for SID retrieval used in our study is an improvement of the traditional MCC algorithm. In our study, the reliability and accuracy of the algorithm are verified via comparison with the buoy. Additionally, the multi-template matching yields satisfactorily in terms of computational speed. Thus, we consider that our method for SID retrieval balances the computational cost and accuracy.

Our study verified the feasibility and validity of using CZI images for SID retrieval in the FS. The spatial resolution of the retrieved SID is 4 km, which is much higher than the SID products based on scatterometers, radiometers, and AVHRR. For our result, the spatial resolution is suitable for capturing complex sea ice motion in the FS. The SID products from scatterometers and radiometers, which possess broader coverage, are more suitable for the study of sea ice mass balance and assimilation of numerical prediction over the whole Arctic region. Our result also illustrates the high potential of the CZI for exploring the polar region. At present, the HY-1C and HY-1D satellites are networked for observation, and the Chinese HY-1 series of satellites will be further utilized to advance polar research. Since 23 December 2021, the discontinuation of the Sentinel-1B satellite operation has resulted in numerous data gaps in the CMEMS SID product, which severely limited the long-term sea ice motion study in the FS. Through leveraging expansions of HY1-C data in future, we anticipate that SID retrieved from CZI will serve as the complement for the CMEMS SID product and facilitate the utilization of high spatial and temporal resolution product for sea ice dynamics research.

*Data availability.* The HY-1D data are available at https://osdds.nsoas.org.cn/DataRetrieval (last access: 26 December 2023, NSOAS, 2023). If interested parties have not registered before, then an account will need to be created to access the data. Upon registration, an account login and password will enable access to the official

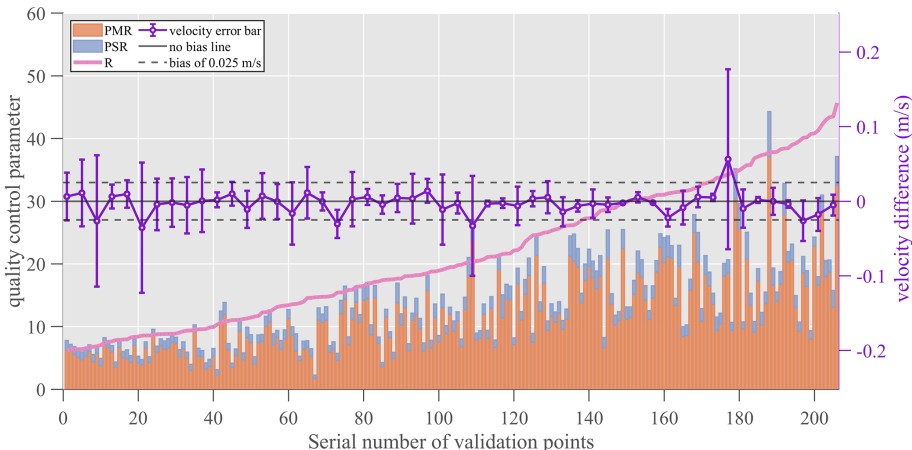

**Figure 21.** Relationship between the velocity difference and quality control parameters.

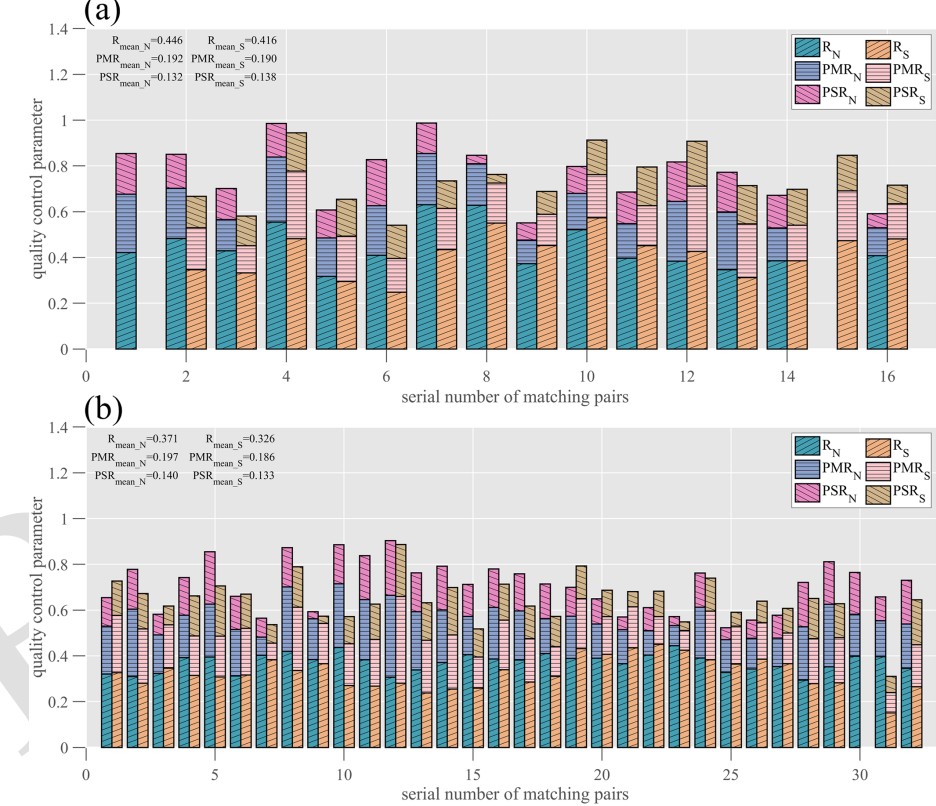

**Figure 22.** Statistics of quality control parameters for individual retrieval SID (**a**, hour level; **b**, day level). The suffixes N and S mean north and south of 80° N in the FS.

website and the HY1-D data. The sea ice drift product is available at https://doi.org/10.48670/MOI-00135, and the product was provided by the Copernicus Marine Service Information (CMEMS) (European Union–Copernicus Marine Service, 2015). The IABP buoy data are available at https://iabp.apl.uw.edu/Data_Products/BUOY_DATA/FULL_RESOLUTION_DATA/Arctic/, provided by the International Arctic Buoy Programme (TS8).

*Author contributions.* Data curation: DL and LS. Writing: DL and LS. Methodology: DL, LS, JL, TZ, BC, SW, and MW. Validation: DL, TZ, and SW. Funding acquisition: LS and JL. All authors have read and agreed to the published version of the paper. JL played a key role in HY satellite technical configuration and made contributions during the preparation of this article.

*Competing interests.* At least one of the (co-)authors is a member of the editorial board of *The Cryosphere*. The peer-review process was guided by an independent editor, and the authors also have no other competing interests to declare.

ther geographical representation in this paper. While Copernicus Publications makes every effort to include appropriate place names, the final responsibility lies with the authors.

*Acknowledgements.* The authors would like to thank the editors and reviewers for their invaluable efforts in improving the paper. The authors would also like to thank the National Satellite Ocean Application Service (NSOAS), CMEMS, DTU, and IABP for providing all the data needed for this paper. The authors would also like to thank the GIV CE5 creators and PIVlab crew for providing referential code.

We dedicate this study to Jianqiang Liu, who sadly passed away in the line of work on 25 May 2023.

*Financial support.* This research has been supported by the National Key Research and Development Program of China (grant nos. 2018YFC1407200 and 2021YFC2803300). TS9

The research has been funded by the National Key Research and Development Program of China (grant nos. 2021YFC2803300 and 2018YFC1407200), the Impact and Response of Antarctic Seas to Climate Change programme (grant no. IRASCC2020-2022-No. 01-01-03 CE6). Bin Cheng has been supported by the European Union's Horizon 2020 research and innovation framework programme (PolarRES project; grant no. 101003590).

*Review statement.* This paper was edited by Xichen Li and reviewed by two anonymous referees.

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

## Remarks from the language copy-editor

## Remarks from the typesetter