# Peer review of "Retrieval of sea ice drift in the Fram Strait based on data from Chinese satellite HaiYang (HY1-D)"

_EGUsphere, 2023_

## Referee Comment (RC2)

**Review on "Retrieval of sea ice drift in the Fram Strait based on data from Chinese satellite HaiYang (HY1-D)"**

The Arctic sea ice reacts strongly to climate change. The common and well-recognized features are thinning of sea ice thickness and shrinking of sea ice extent. As a consequence, the ice drift pattern may differ significantly during the past few decades. The Fram Strait is an important pathway for the sea ice export. For this reason, sea ice drift in the Fram Strait is retrieved based on improved method of the template matching and the Coastal Zone Imager (CZI) imagery from HaiYang-1D (HY-1D) satellite in this study. The authors illustrated the reliability of the product and the spatial and temporal variability of the sea ice motion by comparing with the IABP buoy and CMEMS SAR products. I see it is important and necessary to have different satellite products added to the pool of Arctic data archive. High-resolution sea ice drift product is valuable for the study of sea ice kinematic and deformation, and sea ice export through the Fram Strait is very pertinent to study sea ice changes. For these reasons, I would like to see this manuscript published in TC. However, I see further improvements of this manuscript are necessary in order to warrant acceptance. Please see my comments below and I hope authors can make improvement accordingly.

**General comments:**

The authors used the IABP buoy to validate the product, but due to lacking of on-site observation data, there are uncertainties for the quality of product. The quality control is used to ensure the quality of product and the distribution of these parameters magnitude requires discussion. The spatial resolution of retrieved sea ice drift is different with those of the IABP buoy and CMEMS SAR products. How the authors process the problems during the comparisons between them. Additionally, the authors should illustrate the uncertainties for CMEMS SAR products and IAMP buoy which will bring biases for comparative results. I suggest the authors to carefully proofread the manuscript and resolve all language issues, as it would be very difficult to

pin-point all such issues.

**Special comments:**

Line 16: "has been retrieving" should be "has been retrieved"

Line 37-38: "and it is the process of sea ice as it moves across the sea surface in response to winds, currents and other forces"- Rephrase this statement.

Line 39: the word "consist of" is not quite appropriate. Please recheck it

Line 40: "the TPD" should be "TPD"

Line 51: "With the launch of many remote sensing satellites…" this statement is too colloquial and rewrite it.

Line 55: "Products" should be "products"

Line 56: "…yield lower-resolution due to"- Please recheck this sentence.

Line 57: "OSI SAF scatterometer and radiometer based on SID products are available for many years…" should be "OSI SAF SID products based on scatterometer and radiometer are …"-Please recheck it.

Line 63: the word "geo-parameter retrieval" is not quite appropriate here.

Line 65: "optical remote sensing data" should rewrite as optical imagery, check the whole manuscript.

Line 76-77: the sentence of "but the defect in which feature tracking cannot produce vectors…" is hardly comprehended. Please rephrase this sentence.

Line 80-81: "However, it has been observed that the accuracy of the SID product with AVHRR is not good in s regions…"-Recheck this sentence

Line 91-93: "Multiyear ice (MYI) drift…exist as drift ice"-Recheck this sentence.

Line 95-99: "For our study, in comparison with other products, the retrieved SID from CZI images achieves good accuracy in the FS…" can be as the conclusions and should not put it here.

Line 98: "a sophisticated method was needed to retrieve the motion of drift ice", the sophisticated method should have references.

Line 98: "The data enhancement process can aid our algorithm", the word "aid" is not quite appropriate here.

Line 110: "The wide swath and high resolution of CZI give us an opportunity to understand the sea ice motion in the FS in detail", the sentence is colloquial and need to refine.

Line 132-133: How about the validated result? it should be provided in the manuscript.

Line 134-135: "The CMEMS product with more overlay…"-Recheck this sentence.

Line 250: "The utilization of correlation coefficients and their derived parametric filtering and neighborhood filtering enhances the quality of the results", the word "enhances" is not quite appropriate here.

Line 259: In this study, higher-resolution SID fields are retrieved using CZI with a resolution of approximately 4 km while the grid resolution of the CMEMS SID product is 10 km. How the authors process the discrepancy of spatial resolution for the two products during the comparisons?

Line 271: "…recovered SID…" – is it "retrieved SID"?

Line 283-284: "In our study, a small template is chosen considering the retrieved…less than 0.25m/s"-why a small template will result in this result, authors can illustrate the reasons.

Line 320-321: "an RMSE" should be "a RMSE"

Line 336: "combined with Table 4 and Table 5"-is it "Table 3 and Table 4"?

Line 391: Figure 14 shows the time interval of SAR images used for the CMEMS SID products while the legend in figure 14 shows day -level (CZI)? Please recheck it.

The red line in Figure 18 is unnecessary.

Line 438: "As seen from the mean values, each quality … than for …"-the statement is incomplete.

Line 515: "Our method using the multi-template matching and subpixel estimation approach to retrieve SID in the FS produces a promising result. "a promising result" is not quite appropriate here.

**Other comments:**

The differences of quality control parameters in different regions of Fram Strait should be explained.

The paper should introduce more detailed information about HaiYang series satellites and the level of data.

The relationship between drift distance and velocity retrieval accuracy should be explained.

Whether the time interval of images is appropriate for mosaicking images?

The study explored how the distance of sea ice motion affects the accuracy of flow direction retrieval, but the theory still not clear yet.

---

## Author Comment (AC1)

Dear reviewer

Thank you for your review comments. Based on the evaluations, we have made a major revision of our manuscript:

1, We analyze the effectiveness and spatial distribution of quality control parameters in section 5.6.

2, We added the necessary description of the uncertainties of the CMEMS SID product and IABP buoys in section 2.

3, We have fixed the problems in Fig. 14 and Fig. 18.

4, We carefully checked the language of the manuscript and made revisions

Please see below our response (blue text) to your comments (black text) point-by-point. We have carefully reviewed and addressed all of comments which we hope meet with approval.

Thank you for your time and help,

Best regards,

Dunwang Lu and co-authors

**Responses to Reviewer's Comments:**

**Reviewer #2:**

**General comments**

• GC: "The authors used the IABP buoy to validate the product, but due to lacking of on-site observation data, there are uncertainties for the quality of product. The quality control is used to ensure the quality of product and the distribution of these parameters magnitude requires discussion. The spatial resolution of sea ice drift is different with those of the IABP buoy and CMEMS SAR products. How the authors process the problems during the comparisons between them. Additionally, the authors should illustrate the uncertainties for CMEMS SAR products and IABP buoy which will bring biases for comparative results. I suggest the authors to carefully proofread the manuscript and resolve all language issues, as it would be very difficult to pin-point all such issues."

**Response:** Thank you for your valuable comments. According to your comments, we modified our manuscript in the following four aspects:

**(1) Uncertainty of the validation data**

For the uncertainty of the CMEMS SID product, we check the user manual and find the validation of the product with ITP buoys. The number of matched pair is 29180, the correlation coefficient between the product and ITP buoys is 0.99, and RMSD of dx and dy is 362.32 m (0.0042 m/s) and 339.81 m (0.0039 m/s), the BIAS of dx and dy is 4.64 m and 17.29 m and the BIAS of velocity is negligible (~0 m/s). The time of validation is 2021 and the validation is performed with the 24-hour mean composite product. The website for the

validation report of the product is http://www.seaice.dk/Copernicus/validation. The validation from the following table proves that the product presents high accuracy.

**Table: Latest validation results of the CMEMS SID product.**

| year | metric | coordinate | Value/[m] |
|---|---|---|---|
| 2021 | RMSD, mean | X | 362 |
| | | Y | 340 |
| | BIAS, mean | X | 5 |
| | | Y | 17 |
| 2022 | RMSD, mean | X | 298 |
| | | Y | 402 |
| | BIAS, mean | X | -2 |
| | | Y | -6 |
| 2023 | RMSD, mean | X | 223 |
| | | Y | 236 |
| | BIAS, mean | X | -7 |
| | | Y | -3 |

As for the uncertainty of buoys, we used two kinds of buoys in validation. The GPS position of MOSAiC buoys has an accuracy of ± 2.5 m (Qiu and Li, 2022) which is sufficiently ideal for retrieved SID validation. As for IABP buoy, the buoy positions have an accuracy about 300 m(Haarpaintner, 2006), which is negligible relative to the SID resolution of 4 km.

The necessary descriptions about the uncertainties of the CMEMS SID product and IABP buoys have been added to the section 2.

**(2) Comparison method with buoy data and SAR data**

About the comparison with the buoy, we selected the adjacent buoys in the same SID grid (4km) firstly, then calculated the mean value of the selected buoys (Lavergne et al., 2021; Hwang, 2013; Lavergne et al., 2010).

For the comparison with the CMEMS SID product, the spatial resolution of the product is 10 km and the spatial resolution of retrieved SID is 4 km. The retrieved result was resampled from 4km into 10 km with linear interpolation. The necessary descriptions of comparison have been added to the manuscript.

**(3) Quality control parameters**

To evaluate the quality of SID, we analyzed the effectiveness of the quality control parameters. The following text was added to the manuscript.

To validate the effectiveness of quality control parameters, a comprehensive examination was undertaken, involving a comparative analysis between the quality control parameters and buoy validation of the SID. The red line in Fig. 21 shows the R of the validation points, and to identify the relationships between R and the other parameters, the data are sorted in ascending order by R. The bar in orange and bar in blue represent the PMR and PSR, respectively. The figure revealed a pronounced congruence, wherein the R, PMR and PSR exhibited coherent consistency. A positive correlation was discerned, indicating that the increase in R proportionally accompanies the increase in the PMR and PSR. The magnitude of R has been extensively used as a quality control parameter for SID retrieval (Qiu and

Li, 2022; Haarpaintner, 2006; Robert Ezraty et al., 2007). Given the observed synchronous trends of the PMR, PSR and R, it is evident that the PMR and PSR collectively demonstrate efficacy in quality control. In Fig. 21, the error-bar in purple shows the difference between the retrieved SID and buoys. The bias and standard deviation of the retrieved SID diminish as R, the PMR and the PSR increase, which indicates the effectiveness of these parameters for revealing the reliability of the result.

[Figure]

**Figure 21: Relationship between the velocity difference and quality control parameters.**

**(4) Language issues**
We have carefully reviewed the manuscript and made revisions. The revisions are marked in the manuscript.

**Specific comments:**

**Line 16: "has been retrieving" should be "has been retrieved"**

**Response:** Thanks for the valuable comment, we have fixed the mistake.

**Line 37-38: "and it is the process of sea ice as it moves across the sea surface in response to winds, currents and other forces"- Rephrase this statement.**

**Response:** Thank you for the valuable comments. We rewrote the sentence as 'Sea ice drift (SID) is an important geophysical parameter to describe the dynamic of sea ice and the sea ice motion under the influence of winds, currents, and various external forces'.

**Line 39: the word "consist of" is not quite appropriate. Please recheck it**

**Response:** Thank you for the valuable comment. We rewrote the sentence as 'The primary SID circulation across the Arctic encompasses both the Beaufort Gyre (BG) and the Transpolar Drift (TPD)'.

**Line 40: "the TPD" should be "TPD"**

**Response:** Thanks for the valuable comment, we have fixed the mistake.

**Line 51: "With the launch of many remote sensing satellites…" this statement is too colloquial and rewrite it.**

**Response:** Thank you for the valuable comment. We rewrote the sentence as 'The development of satellites and remote sensing sensors promote satellite data as a prevailing trend in retrieving SID'.

**Line 55: "Products" should be "products"**

**Response:** Thanks for the valuable comment, we have fixed the mistake.

**Line 56: "…yield lower-resolution due to"- Please recheck this sentence.**

**Response:** Thank you for the valuable comment. We rewrote the sentence as 'SID products derived from radiometers and scatterometers inherently possess coarse spatial resolution owing to the characteristics of the sensors.'

**Line 57: "OSI SAF scatterometer and radiometer based on SID products are available for many years…" should be "OSI SAF SID products based on scatterometer and radiometer are …"-Please recheck it.**

**Response:** Thank you for the valuable comment. We rewrote the sentence as 'OSI SAF provides SID products retrieved from scatterometers and radiometers over the polar regions and its temporal coverage is from 2009 to now'.

**Line 63: the word "geo-parameter retrieval" is not quite appropriate here.**

**Response:** Thank you for the valuable comment. We rewrote the sentence as 'Optical imagery has been applied extensively in cryosphere observation'.

**Line 65: "optical remote sensing data" should rewrite as optical imagery, check the whole manuscript.**

**Response:** Thanks for the valuable comment, we have fixed the mistake.

**Line 76-77: the sentence of "but the defect in which feature tracking cannot produce vectors…" is hardly comprehended. Please rephrase this sentence.**

**Response:** Thank you for the valuable comment. We rewrote the sentence as 'The accuracy of their result is promising (Fang et al., 2023), but the spatial coverage needs to be further improved'.

**Line 80-81: "However, it has been observed that the accuracy of the SID product with AVHRR is not good in s regions..."-Recheck this sentence**

**Response:** Thank you for the valuable comment. We rewrote the sentence as 'However, it has been found that the accuracy of the SID product retrieved from AVHRR presents low accuracy in East Greenland, with the Mean Absolute Error (MAE) of velocity reaching 10.40 km/day, which is even lower than that of the SID products retrieved from radiometer and scatterometer (Wang et al., 2022)'.

**Line 91-93: "Multiyear ice (MYI) drift...exist as drift ice"-Recheck this sentence.**

**Response:** Thank you for the valuable comment. We rewrote the sentence as 'The multiyear ice (MYI) drifts from the Arctic basin and crushes in the central part of strait, which results in the fragmented ice in the southern part of the FS and along the eastern coast of Greenland'.

**Line 95-99: "For our study, in comparison with other products, the retrieved SID from CZI images achieves good accuracy in the FS..." can be as the conclusions and do not put it here.**

**Response:** Thank you for the valuable comment, we deleted these sentences in the introduction.

**Line 98: "a sophisticated method was needed to retrieve the motion of drift ice", the sophisticated method should have references.**

**Response:** Thank you for the valuable comment, we deleted the sentence to make the manuscript more readable.

**Line 98: "The data enhancement process can aid our algorithm", the word "aid" is not quite appropriate here.**

**Response:** Thank you for the valuable comment, we deleted the sentence due to inappropriate expression.

**Line 110: "The wide swath and high resolution of CZI give us an opportunity to understand the sea ice motion in the FS in detail", the sentence is colloquial and need to refine.**

**Response:** Thank you for the valuable comment. We rewrote the sentence as 'The wide swath and high spatial resolution of CZI imagery make it suitable for the sea ice motion observation in FS'.

**Line 132-133: How about the validated result? it should be provided in the manuscript.**

**Response:** Thank you for the valuable comment. For the uncertainty of SID product, we

check the CMEMS SID product user manual and find the validation of the product with ITP buoys. The number of matched pair is 29180, the correlation coefficient between the product and ITP buoys is 0.99, and RMSD of dx and dy is 362.32 m (0.0042 m/s) and 339.81 m (0.0039 m/s), the BIAS of dx and dy is 4.64 m and 17.29 m and the BIAS of velocity is negligible (~0 m/s). The time of validation is 2021 and the validation is performed with the 24-hour mean composite product. The validation shows very good correlation (almost 1.0) when tracking offsets are compared to a validation reference. Mean error values are very small, which indicates almost no bias, and the standard deviation of the differences is low. The validation proves that the CMEMS SID product has great accuracy. We added necessary descriptions about the validation in Section 2.2.

**Line 134-135: "The CMEMS product with more overlay…"-Recheck this sentence.**

**Response:** Thank you for the valuable comment. For the CMEMS SID product, the composite product is updated every 12 hours covering 24 hours. Thus, two kinds of product are provided, with a nominal time interval of 0:00 to 0:00 and 12:00 to 12:00, respectively. we rewrote the sentence as 'Therefore, the CMEMS SID product which has the most temporal overlap with CZI images is chosen for comparison'. In addition, we have added more descriptions of the CMEMS SID product.

**Line 250: "The utilization of correlation coefficients and their derived parametric filtering and neighborhood filtering enhances the quality of the results", the word "enhances" is not quite appropriate here.**

**Response:** Thank you for the valuable comment, we deleted the sentence due to inappropriate expression.

**Line 259: In this study, higher-resolution SID fields are retrieved using CZI with a resolution of approximately 4 km while the grid resolution of the CMEMS SID product is 10 km. How the authors process the discrepancy of spatial resolution for the two products during the comparisons?**

**Response:** Thanks for the valuable comment. We resampled our SID product from 4 km into 10 km with linear interpolation method. By resampling, our retrieval SID had the same spatial resolution as the CMEMS SID product. The necessary descriptions were added in the section 2.2.

**Line 271: "…recovered SID…" – is it "retrieved SID"?**

**Response:** Thanks for the valuable comment, we have changed the 'recovered' to 'retrieved'.

**Line 283-284: "In our study, a small template is chosen considering the retrieved…less than 0.25m/s"-why a small template will result in this result, authors can illustrate the reasons.**

**Response:** Thanks for the valuable comment. The size of the template and search area is associated with the spatial resolution of result. During the template matching, larger template size and search area means that the algorithm can find the maximum correlation point in wider area, but also it will lead to low-resolution result. The effect of template size and search area on retrieving SID will be explored in future research. We've rewritten the paragraph to make the expression clear. The following text was added to the manuscript.

The size of the template and search area is associated with the spatial resolution of result. To retrieve high spatial resolution SID in the FS, limited template size and search area is set in our study. Therefore, the maximum velocity of day-level result is lightly smaller than the product. The effect of template size and search area on retrieving SID will be explored in future research.

**Line 320-321: "an RMSE" should be "a RMSE"**

**Response:** Thanks for the valuable comment, we have changed all 'an RMSE' to 'a RMSE'.

**Line 336: "combined with Table 4 and Table 5"-is it "Table 3 and Table 4"?**

**Response:** Thanks for the valuable comment, we have fixed the mistake.

**Line 391: Figure 14 shows the time interval of SAR images used for the CMEMS SID products while the legend in figure 14 shows day -level (CZI)? Please recheck it.**

**Response:** Thanks for the valuable comment. In our study, the CZI dataset was divided into two different categories: hours-level and day-level. During comparing the result with the CMEMS SID product, discrepancies were identified with the hours-level result. The Figure 14 shows the corresponding time intervals of the CMEMS SID product in comparison and there is no obvious difference in the time interval between SAR images of the products used to compare with our results. The legend was modified to express our opinion clearly.

[Figure]

**Figure 14: The time interval of SAR images used by the CMEMS SID product during the comparison with two different results.**

**The red line in Figure 18 is unnecessary.**

**Response:** Thanks for the valuable comment, we deleted the red line.

**Line 438: "As seen from the mean values, each quality … than for …"-the statement is incomplete.**

**Response:** Thanks for the valuable comment, we have fixed the mistake. We rewrote the sentence as 'Each quality control parameter's mean value of the hours-level result is greater than that of the day-level result, which indicates that the hours-level result has the higher quality'.

**Line 515: "Our method using the multi-template matching and subpixel estimation approach to retrieve SID in the FS produces a promising result. "a promising result" is not quite appropriate here.**

**Response:** Thanks for the valuable comment, we rewrote the sentence as 'Our method, utilizing the multi-template matching and subpixel estimation approach for retrieving SID in FS, demonstrates great accuracy'.

**Other comments:**

**The differences of quality control parameters in different regions of Fram Strait should be explained.**

**Response:** Thanks for the valuable comment. The Fram Strait, where the sea ice changes dramatically, the sea ice in the southern and northern of the strait render different

morphologies in the images. Investigating the quality of retrieved SID in different regions associated with quality control parameters is valuable. The following text was added to the section 5.6 to discuss the quality control parameters in different regions.

The coherence between the quality control parameters and the validation of the buoys is illustrated in Fig. 21, providing a foundation for the meticulous examination of the SID quality. As one of the main outlets of Arctic sea ice in the Atlantic Ocean, the sea ice concentration in the FS is inferior than the Arctic Basin (Peng and Meier, 2018; Wang et al., 2020), and the sea ice in the south and north of the strait render different morphologies in the images.

To investigate the quality of SID over the strait, we selected 80°N as the segmentation line and calculated the quality control parameters of SID in north and south of the segmentation line. The Fig. 22 shows the stacking bars of different parameters in different regions. For the sake of visualization, we normalize the PMR and PSR. The average values of these quality control parameters indicate that SID in the north of 80° N have higher quality. Comparison of quality control parameters for results with different time intervals reveals that the hours-level result possesses higher quality. Besides, the hours-level result also has greater mean value of quality control parameters in the south of 80° N where the sea ice is dispersive. High-concentration sea ice constrains the variability of sea ice motion and provides better spatial consistency, which is beneficial for retrieving SID. Similarly, the variability of the sea ice motion with short time interval is inconspicuous, providing a favorable scene for the algorithm. The sea ice kinematics in marginal ice zone are intricate, and the effects of wave on the fragmentary sea ice motion are evident (Williams et al., 2013, p.1), which increases the uncertainty of retrieval. The utilization of short time interval images for SID retrieval proves instrumental in enhancing the quality of monitoring sea ice motion within marginal ice zones.

[Figure]

**Figure 22: Statistics of quality control parameters for individual retrieval SID. (a: hours-level, b: day-level). The suffix N and S mean north and south of 80°N in FS.**

**The paper should introduce more detailed information about HaiYang series satellites and the level of data.**

**Response:** Thanks for the valuable comment. The following text was added to the section 2.1 to introduce the Haiyang satellites and the level of data.

Since the launch of China's first ocean satellite in 2002, concerted efforts have been undertaken to institute a comprehensive global operational ocean satellite observation system. Currently, the observation system consists of 10 satellites, which include three series: ocean color series satellites (HY-1), ocean dynamic environment series satellites (HY-2), and ocean surveillance and monitoring series satellites (HY-3) (Zeng et al., 2023). In this paper, SID retrieval is performed using the L1C data which is processed with radiometric calibration and geographic projection. Before retrieving, the CZI images are resampled to 300 m considering the algorithm's computational efficiency and the spatial resolution of the result.

**The relationship between drift distance and velocity retrieval accuracy should be explained.**

**Response:** Thanks for the valuable comment. It is necessary to explore the relationship between the accuracy of the retrieval velocity and the distance of sea ice motion. We thus add a new section '5.5 Factors affecting the accuracy of velocity retrieval' to discuss how the distance and velocity of sea ice motion affect the retrieval accuracy of velocity in our study. The following text was added to the section 5.5.

Additionally, we delve into the correlation between the distance of sea ice motion and the accuracy of the retrieved velocity. A significant challenge to the accuracy of velocity retrieval, as emphasized by Lavergne, is the presence of quantification noise in template matching (Lavergne et al., 2010). Despite the acknowledged presence of quantification noise, our result surprisingly does not reveal its discernible effects, and a lack of a statistically significant correlation between the accuracy of the retrieved velocity and buoy distance is observed (As shown in Fig. 20).

Several explanations underpin this result. First, the spatial resolution of the resampled CZI imagery is 300m, which is a notable improvement for the ice surface feature observation compared to the kilometer-level resolution of radiometer. The higher spatial resolution is posited to be a key factor limiting the manifestation of quantification noise. The topographic features of sea ice are often quantified by the surface roughness and form drag (Arya, 1973, 1975) and those observed surface features have impact on ice drift speed (Zu et al., 2021). Unfortunately, the CZI imagery and resampled images are inadequate in detecting the topographic features. Second, the strong sea ice motion in FS renders sea ice dynamics distinctly visible in the image, indirectly suppressing the appearance of quantification noise. Finally, our application of subpixel estimation for the precise

localization of maximum correlation values, results in an exact determination of the maximum correlation location. This refinement contributes to improve the accuracy of retrieving SID. In conclusion, high spatial resolution data and subpixel estimation are able to suppress the negative effects caused by quantification noise.

[Figure]

**Figure 20: Relationship between the velocity difference and the displacement of the buoys.**

**Whether the time interval of images is appropriate for mosaicking images?**

**Response:** Thanks for the valuable comment. CZI images from the same orbit are used for SID retrieval. Besides, we used images from different moments to find the intersection region and crop. Actually, we generated the dataset without mosaicking. We made necessary changes to the flowchart to correct the mistake in expression.

**The study explored how the distance of sea ice motion affects the accuracy of flow direction retrieval, but the theory still not clear yet.**

**Response:** Thanks for the valuable comment. The following text was added to the section 5.4 to illustrate how drift distance affects flow direction retrieval.

Due to stochastic error, the identification of the highest correlation point deviates to the optimal point. Consequently, the retrieved flow direction is constrained to the vicinity of the optimal point. The variability of flow direction retrieval with longer distance is more stable than with shorter distance.

As illustrated in the following figure, sea ice motion with longer distance (the blue vector) demonstrates better accuracy in flow direction retrieval compared to motion with shorter distance (the wheat vector).

[Figure]

**Figure: Illustration of how distance affects flow direction retrieval. Vectors with different color show the sea ice motion with different drift distance (wheat vector represent SID with short distance, blue vector with long distance), the square with color is the potential retrieval value, the small red curve is the potential range of retrieved flow direction.**

**Reference:**

Arya, S. P. S.: Contribution of form drag on pressure ridges to the air stress on Arctic ice, Journal of Geophysical Research (1896-1977), 78, 7092–7099, https://doi.org/10.1029/JC078i030p07092, 1973.

Arya, S. P. S.: A drag partition theory for determining the large-scale roughness parameter and wind stress on the Arctic pack ice, Journal of Geophysical Research (1896-1977), 80, 3447–3454, https://doi.org/10.1029/JC080i024p03447, 1975.

Fang, Y., Wang, X., Li, G., Chen, Z., Hui, F., and Cheng, X.: Arctic sea ice drift fields extraction based on feature tracking to MODIS imagery, International Journal of Applied Earth Observation and Geoinformation, 120, 103353, https://doi.org/10.1016/j.jag.2023.103353, 2023.

Haarpaintner, J.: Arctic-wide operational sea ice drift from enhanced-resolution QuikScat/SeaWinds scatterometry and its validation, IEEE Trans. Geosci. Remote Sensing, 44, 102–107, https://doi.org/10.1109/TGRS.2005.859352, 2006.

Hwang, B.: Inter-comparison of satellite sea ice motion with drifting buoy data, International Journal of Remote Sensing, 34, 8741–8763, https://doi.org/10.1080/01431161.2013.848309, 2013.

Lavergne, T., Eastwood, S., Teffah, Z., Schyberg, H., and Breivik, L.-A.: Sea ice motion from low-resolution satellite sensors: An alternative method and its validation in the Arctic, Journal of Geophysical Research: Oceans, 115, https://doi.org/10.1029/2009JC005958, 2010.

Lavergne, T., Piñol Solé, M., Down, E., and Donlon, C.: Towards a swath-to-swath sea-ice drift product for the Copernicus Imaging Microwave Radiometer mission, The Cryosphere, 15, 3681–3698, https://doi.org/10.5194/tc-15-3681-2021, 2021.

Peng G. and Meier W. N.: Temporal and regional variability of Arctic sea-ice coverage from satellite data, Annals of Glaciology, 59, 191–200, https://doi.org/10.1017/aog.2017.32, 2018.

Qiu, Y. and Li, X.-M.: Retrieval of sea ice drift from the central Arctic to the Fram Strait based on sequential Sentinel-1 SAR data, IEEE Trans. Geosci. Remote Sensing, 1–1, https://doi.org/10.1109/TGRS.2022.3226223, 2022.

Robert Ezraty, Fanny Girard-Ardhuin, and Croizé-Fillon: Sea-Ice Drift in the Central Arctic Using the 89 GHz Brightness Temperatures of the Advanced Microwave Scanning Radiometer, 2007.

Wang, X., Chen, R., Li, C., Chen, Z., Hui, F., and Cheng, X.: An Intercomparison of Satellite Derived Arctic Sea Ice Motion Products, Remote Sensing, 14, 1261, https://doi.org/10.3390/rs14051261, 2022.

Wang, Z., Li, Z., Zeng, J., Liang, S., Zhang, P., Tang, F., Chen, S., and Ma, X.: Spatial and Temporal Variations of Arctic Sea Ice From 2002 to 2017, Earth and Space Science, 7, e2020EA001278, https://doi.org/10.1029/2020EA001278, 2020.

Williams, T. D., Bennetts, L. G., Squire, V. A., Dumont, D., and Bertino, L.: Wave–ice interactions in the marginal ice zone. Part 1: Theoretical foundations, Ocean Modelling, 71, 81–91, https://doi.org/10.1016/j.ocemod.2013.05.010, 2013.

Zeng, T., Shi, L., Huang, L., Zhang, Y., Zhu, H., and Yang, X.: A Color Matching Method for Mosaic HY-1 Satellite Images in Antarctica, Remote Sensing, 15, 4399, https://doi.org/10.3390/rs15184399, 2023.

Zu, Y., Lu, P., Leppäranta, M., Cheng, B., and Li, Z.: On the Form Drag Coefficient Under Ridged Ice: Laboratory Experiments and Numerical Simulations From Ideal Scaling to Deep Water, Journal of Geophysical Research: Oceans, 126, e2020JC016976, https://doi.org/10.1029/2020JC016976, 2021.

---

## Author Comment (AC2)

Dear reviewer,

Thank you for your comments. Based on the evaluations, we have made a major revision of our manuscript as below:

1, We added quantitative analysis on the impact of the cloud in section 5.3.

2, We analyzed the retrieval accuracy of different sea ice velocity in section 5.5.

3, We checked the buoy validation issue and made the necessary modifications in section 4.2.

Please see below our response (blue text) to your comments (black text) point-by-point. We have carefully reviewed and addressed all of comments which we hope meet with approval.

Thank you for your time and help,

Best regards,

Dunwang Lu and co-authors

**Responses to Reviewer's Comments:**

**Reviewer #1:**

**General comments**

• GC: "The biggest problem with optical remote sensing is the impact of clouds. Although the paper has discussed the impact of clouds on sea ice motion products, the extent of the impact and its impact on the effective data are not very clear. Further clarification is needed. In addition, it is also necessary to consider whether the topographic features of summer sea ice surfaces, such as snow hummocks and ice ridges, have an impact on the inversion results. The impact of sea ice motion speed itself on the errors of data product needs to be further quantified, and the spatial and seasonal differences in retrieval errors also need to be quantitatively explained. At present, the paper mainly uses examples to illustrate the above issues, rather than providing statistical results, which is not conducive to objective evaluation of the data product."

Response: Thank you for your valuable comments. According to your comments, we modified our manuscript in the following four aspects:

(1) The impact of clouds on SID products

The cloud is the main drawbacks for the SID retrieval with optical imagery and it effect the coverage of the SID product. At present, the SID based on AVHRR is the only SID product retrieved from optical imagery (e.g., Wang et al., 2022). In our study, the sea ice under thin cloud can be detected with appropriately preprocessing, which increase the coverage of our product. We used whiteness index (Gómez-Chova et al., 2008) and MED (Multiscale Edge-preserving Decompositions) (Farbman et al., 2008) to detect the cloud pixel (Kang et al., 2018) and to analyze the effect of cloud quantitatively. The following text and figure have been added into the section 5.3 of our manuscript as "The potential of retrieving sea

ice drift from image with cloud".

[Figure]

**Figure 17: The number of no-cloud pixels (green) and retrieved SID pixels (blue) (a: day-level result, b: hours-level result).**

To quantify the potential of retrieving SID from image with cloud, we use the whiteness index (Gómez-Chova et al., 2008) and Multiscale Edge-preserving Decompositions (MED) (Farbman et al., 2008) to detect the cloud pixels (Kang et al., 2018). Fig .17(a) and (b) show the number of no-cloud pixels and the number of retrieved SID pixels. For the day-level results, in addition to the influence of cloud pixels, dramatic shape changes of sea ice also hinder the retrieval of SID. As shown in Fig. 17(a), only a few (1st, 15th, 21st, 30th, 31st and 32nd) of the retrieved SID pixels are beyond the no-cloud pixels. For the 32nd image pair, the number of SID pixels is 120% greater than the number of no-cloud pixels, possibly because the cloud in the images is quite thin. For the hours-level results, images with short time interval provide a stable scene for retrieving. Thus, many of the retrieved SID pixels (1st, 6th, 7th, 9th and 11th-16th) are greater than the no-cloud pixels as shown in Fig. 17(b). For the 16th image pair, the number of retrieved SID pixels is 150% greater than the number of no-cloud pixels, this is because the cloud is slight and the distributions of cloud in two images are similar. The average of SID pixels beyond the no-cloud pixels is 10.109% and 28.920% for the day-level result and the hours-level result, respectively. In conclusion, our approach has the potential to retrieve SID effectively with images containing cloud pixels. The result shows that with less strict restriction and appropriate preprocessing, images with cloud could be used to obtain credible and dense SID.

**(2) The impact of sea ice surface feature on SID products**
The topographic features of sea ice are often quantified by the surface roughness and form drag (Arya, 1973, 1975) and those features do have impact on ice drift speed (Zu et al., 2021) In this study, the optical imagery of CZI with 300m spatial resolution was used for SID retrieval and this images have the better identification ability for the ice surface feature than those observation with km spatial solution. Some surface topographic features such as snow hummocks and ice ridges can not be identified with optical images. These features can be observed with SAR and laser altimeter and they can associated with the momentum exchange between sea ice and the lower atmosphere forced by wind (Zhang et al., 2024). The impact of topographic features on SID product will be the future improvements to our product. We have added some description to discuss this matter in the last paragraph of section 5.5.

**(3) Quantified error of the sea ice motion speed**

The impact of sea ice velocity on the accuracy of retrieval SID is imperative, so we made a new experiment and the below text has been added in the manuscript as section 5.5.
5.5 Factors affecting the accuracy of velocity retrieval.

[Figure]

**Figure 19: Relationship between the velocity difference and the velocity of the buoys.**

The study area spans the Arctic basin to the Fram Strait, where the velocity of the SID generally presents complicated variations. The maximum magnitude of buoy-derived ice velocity can reach to 0.64 m/s in FS (Lei et al., 2016). Sea ice in the north of the FS or close to the land has a relatively low velocity. It is imperative to make further research on the accuracy of retrieving SID with different velocity.

In our study, the data were from March to May 2021, and we generated a total of 206 matched points and merged every 4 points to calculate the standard deviation of the velocity difference. We further analyzed the relation between the accuracy of retrieved velocity and the buoy velocity. As shown in Fig 19, the bar shows that the velocity of buoys that we used in validation ranged from 0 m/s to 0.32 m/s, and the line of bias and standard deviation shows the accuracy of the retrieved velocity which is associated with the velocity of the buoys. As the velocity of the buoys increases to 0.1 m/s, both the bias and standard deviation also increase. Although the bias of retrieved velocity occurs within in an ideal range as the velocity of the buoys increases, the standard deviation of the velocity difference becomes erratic. In general, sea ice motion with great velocity appears unstable in the images. Our algorithm still need improvement in retrieving SID with high velocity.

**(4) The spatial and seasonal distribution of retrieval errors**

The seasonal differences in retrieval errors are important. The images used in this study were collected from March to May 2021, thus we do not have enough data to analyze the seasonal change in the retrieval errors. We are considering processing HY1-C data to improve our dataset in the future experiment.

For the spatial differences in retrieval error, we generated 206 matched points and merged every 4 points to calculate the standard deviation of velocity difference. We plotted the relationship between the velocity difference and the latitude of validation buoys in the following figure, the gray bar in the X-axis refers to the latitude of the buoys. We didn't

observe the significant correlation between spatial differences and retrieval errors. It's challenging for us to explain the difference of spatial retrieval errors on the limited amount of data. Improving our dataset and including more buoys are required in future to explore the retrieval error in different region.

[Figure]

**Figure: Relationship between the velocity difference and the latitude of validation pair.**

**Specific comments:**

**Line 34 "leading to accelerated sea ice break-up" -- ice break-up generally describes the situation of synoptic scale processes.**

**Response:** Thank you for the valuable comment. We rewrite the sentence as "Arctic quick warming accelerates the melting of polar sea ice, leads to thinner sea ice and accelerates sea ice transport (Krumpen et al., 2016; Maslanik et al., 2011)."

**Line 40 "the TPD transports large quantities of multiyear ice outward from the central Arctic toward the FS"-- not just the Fram Strait, but also the Barents Sea and Baffin Bay.**

**Response:** Thank you for the valuable comment. We rewrite the sentence as "TPD transports large quantities of multiyear ice outward from the central Arctic toward the FS, Barents Sea and Baffin Bay (Colony and Thorndike, 1984; Martin and Augstein, 2000)."

**Line 46 "which gradually melts during outward transport"--If the sea ice outflow occurs during winter, sea ice growth may also occur.**

**Response:** Thank you for the valuable comment. The outflow is considered as the sea ice move from the Arctic basin to the low latitude area. We rewrite the sentence as "Sea ice, a mixture of ice and brine (Schwerdtfecer, 1963), which gradually disintegrates during outward transport in the FS." in line 46.

**Line 50 "between the polar regions and the outside world"--what is the meaning of**

**"outside of world"**

**Response:** Thank you for the valuable comment. We rewrite the sentence as "Therefore, observing SID in the FS is crucial to analyse the sea ice variation in the Arctic and the sea ice transport from the polar to sub-polar regions."

**Line 59 "low temporal resolution SID product may fail to provide accurate sea ice drift patterns"-- The main limitation of low-temporal-resolution sea ice motion products is that they cannot depict the subdaily-scale signals of the sea ice kinematics.**

**Response:** Thank you for the valuable comment. We rewrite the sentence as "low temporal resolution SID product may fail to depict the subdaily-scale variation of the sea ice motion."

**Line 81 "However, it has been observed that the accuracy of the SID product with AVHRR is not good in s regions like East Greenland"-- What are the reasons for poor observation results?**

**Response:** Thank you for the valuable comment. We found the conclusion that the accuracy of SID retrieved from AVHRR in east Greenland is poor in Wang's paper (Wang et al., 2022). The reason is not discussed in the paper. We think the low accuracy for the product in east Greenland is due to the disintegrated small sea ice in the FS and low spatial resolution of used images. We rewrite the sentence as "However, it has been found that the accuracy of the SID product retrieved from AVHRR presents low accuracy in East Greenland".

**Line 132 "the product includes the North Pole and South Pole"--1) change to the product is available from both Arctic Ocean and South Ocean; 2) The language of the entire text must be more strictly controlled.**

**Response:** Thank you for the valuable comment. We rewrite the sentence as "and the product is available from both Arctic Ocean and South Ocean (Pedersen et al., 2015).". We checked the language of the entire manuscript and marked our revisions in the manuscript.

**Figure 2: "The drifting trajectories of 69 IABP buoys from March to May 2021": How independent are these data, that is, they are not deployed in a very close area, especially the buoys deployed during the MOSAiC; In addition, whether to conduct quality control on the data and eliminate the data with noise and buoy data that are already at sea (not over the ice).**

**Response:** Thank you for the valuable comment. We had deployed quality control to the buoy data. We checked the location of the buoys data to make sure the buoy is placed on the ice and the corresponding description in Section 2.3 has been modified. There are some buoys are deployed in the vicinity of same SID grid. Thus, the different buoys located in the same SID grid were averaged and used to validate SID. We think that buoys in the same grid are all significative and contributing for validating the SID.

**Line 156 "we design a quality control session to remove the low quality data from the results"-- How much data will be lost during the study period due to the impact of clouds?**

**Response:** Thank you for the valuable comment. Quality control is a vital step for the quality of SID. In our study, we employed cross-correlation parameter (R), peak second ratio (PSR) and peak mean ratio (PMR) to limit quality level of the result. These parameters, especially the cross-correlation parameter, had been widely used in producing SID and also been validated as effective parameters for quality control. About the amount of the filtered data with quality control parameters, we added an explanation in section 5.3.

**The error in the direction of sea ice movement: We know that the sea ice movement in the Fram Strait is relatively stable, so it is possible that the angle error may be small. Can you further evaluate the angle error of sea ice motion under different meandering coefficients?**

**Response:** Thank you for the valuable comment. A greater meander coefficient signifies a more erratic trajectory, whereas a value of M = 1 indicates that the buoy travelled in a straight line(Womack et al., 2022). The meandering coefficient is a primary quantitative measure for the kinematics of sea-ice drift (Vihma et al., 1996). The following figure shows the retrieval angle error and the meandering coefficients of buoys. A total of 306 matched pairs are produced. The number of the matched pairs is less than 344, because that there are some buoys possess less than 3 GPS points which is inadequate for calculating the meandering coefficient. In the meander coefficients, 91.83% of them is less than 1.5. The time interval of our dataset is less than 30 hours, and the meandering coefficient of sea ice motion at short time interval is low relatively. We didn't observe the correlation between the retrieval angle error of sea ice motion and meandering coefficients. It's challenging for us to explain the relation between them because the time interval of our dataset is shorter than one day.

[Figure]

**Figure: The retrieval angle error and the meandering coefficient of buoy.**

**Hourly data: Does the data have the ability to identify the subdaily-scale characteristics of sea ice motion and compare them with buoy data on a frequency basis?**

**Response:** Thank you for the valuable comment. Comparing with the buoys, CZI data has lower frequency in capturing the subdaily-sacle characteristics of sea ice motion. But CZI has wide swath and can provide 3 times observations in one day over a specific area of Arctic. There are more times observation if we combine HY-1C and HY-1D, which supply a good data source for identifying the subdaily-scale characteristics of sea ice motion.

**BIAS: Relative deviation is also very important.**

**Response:** Thank you for the valuable comment. We had added the relative deviation of bias into Table 3 and 4.

**Table 3: Validation of the retrieved SID (day-level) with IABP buoys.**

| Delta | Number of match point | BIAS (Relative deviation$_{mean}$) | MAE | STD | RMSE |
|---|---|---|---|---|---|
| Velocity (m/s) | | -0.005(-1.330%) | 0.018 | 0.031 | 0.031 |
| Flow direction (rad) | 132 | 0.002(0.149%) | 0.003 | 0.009 | 0.009 |

**Table 4: Validation of the retrieved SID (hours-level) with IABP buoys.**

| Delta | Number of match point | BIAS (Relative deviation$_{mean}$) | MAE | STD | RMSE |
|---|---|---|---|---|---|
| Velocity (m/s) | | 0.000(1.998%) | 0.021 | 0.036 | 0.036 |
| Flow direction (rad) | 74 | 0.003(0.154%) | 0.006 | 0.010 | 0.010 |

**Figure 12: What is the confidence level of the correlation coefficient?**

**Response:** Thank you for the valuable comment. The confidence level is an important parameter for correlation coefficient, we had added the 95% confidence level bar in the figure 12, and also added necessary descriptions and analysis for the figure in the manuscript.

[Figure]

**Figure 12: The correlation coefficient and 95% confidence level bar of correlation coefficient between the retrieved SID and the CMEMS SID product. (a), (b): The correlation coefficient between the retrieved SID (day-level) and the product in terms of velocity and flow direction. (c), (d): The correlation coefficient between retrieved SID (hours-level) and the product in terms of velocity and flow direction. The green histogram (right axis) shows the number of matched points between the retrieved SID and the product, and the blue histogram (the first left axis) shows the time interval between the retrieved SID and the product.**

**Figure 15: In the caption of the illustration, a lot of information is missing, which is only appear in the main text.**

**Response:** Thank you for the valuable comment. We rewrite the illustration as "Figure 15: The buoy velocity (a) and flow direction (c) and the difference (velocity (b), flow direction (d)) between the retrieved SID and buoys. The lines in blue represents SID and buoy vectors with day-level time interval and the lines in yellow, gray and purple represents SID and buoy vectors with hours-level time interval on the same day. The x-axis shows the different matched SID and buoys. The y-axis in (b) and (d) show the difference between SID and buoys".

**Figure 16: This is an obvious result, and this illustration is not necessary. It is necessary to add an explanation of the classification evaluation under different conditions with various meandering coefficients and sea ice motion speed. And provide clustering statistical results for different sub regions.**

**Response:** Thank you for the valuable comment. We deleted the figure in the manuscript. The following figure shows the retrieval velocity error and the meandering coefficients of buoys. A total of 306 matched pairs are produced. In the meander coefficients, 91.83% of them is less than 1.5. We didn't observe the correlation between the retrieval velocity error of sea ice motion and meandering coefficients. We set 80°N as the segmentation line and calculated the mean value of meandering coefficients and retrieval errors. In the northern of the segmentation line, the mean of meandering coefficient and velocity error are 1.175, 0.005 m/s, and the number of points is 296. In the southern of the segmentation line, the mean of meandering coefficient and velocity error

are 1.033, -0.010 m/s, and the number of points is 10. A greater meander coefficient signifies a more erratic trajectory. The mean of meander coefficient in the southern of the segmentation line should be bigger than the mean of meander coefficient in the northern of the segmentation line. We think that the reliability of our experiment about meandering coefficient is insufficient because of the limited number of matched pair.

[Figure]

**Figure: The retrieval velocity error and the meandering coefficient of buoy.**

**Reference:**

Arya, S. P. S.: Contribution of form drag on pressure ridges to the air stress on Arctic ice, Journal of Geophysical Research (1896-1977), 78, 7092–7099, https://doi.org/10.1029/JC078i030p07092, 1973.

Arya, S. P. S.: A drag partition theory for determining the large-scale roughness parameter and wind stress on the Arctic pack ice, Journal of Geophysical Research (1896-1977), 80, 3447–3454, https://doi.org/10.1029/JC080i024p03447, 1975.

Farbman, Z., Fattal, R., Lischinski, D., and Szeliski, R.: Edge-preserving decompositions for multi-scale tone and detail manipulation, ACM Trans. Graph., 27, 1–10, https://doi.org/10.1145/1360612.1360666, 2008.

Gómez-Chova, L., Camps-Valls, G., Calpe, J., Guanter, L., and Moreno, J.: Cloud-screening algorithm for ENVISAT/MERIS multispectral images, Geoscience and Remote Sensing, IEEE Transactions on, 45, 4105–4118, https://doi.org/10.1109/TGRS.2007.905312, 2008.

Kang, X., Gao, G., Hao, Q., and Li, S.: A Coarse-to-Fine Method for Cloud Detection in Remote Sensing Images, IEEE Geoscience and Remote Sensing Letters, PP, 1–5, https://doi.org/10.1109/LGRS.2018.2866499, 2018.

Lei, R., Heil, P., Wang, J., Zhang, Z., Li, Q., and Li, N.: Characterization of sea-ice kinematic in the Arctic outflow region using buoy data, Polar Research, 35, 22658,

https://doi.org/10.3402/polar.v35.22658, 2016.

Vihma, T., Launiainen, J., and Uotila, J.: Weddell Sea ice drift: Kinematics and wind forcing, J. Geophys. Res., 101, 18279–18296, https://doi.org/10.1029/96JC01441, 1996.

Wang, X., Chen, R., Li, C., Chen, Z., Hui, F., and Cheng, X.: An Intercomparison of Satellite Derived Arctic Sea Ice Motion Products, Remote Sensing, 14, 1261, https://doi.org/10.3390/rs14051261, 2022.

Womack, A., Vichi, M., Alberello, A., and Toffoli, A.: Atmospheric drivers of a winter-to-spring Lagrangian sea-ice drift in the Eastern Antarctic marginal ice zone, Journal of Glaciology, 68, 999–1013, https://doi.org/10.1017/jog.2022.14, 2022.

Zhang, Z., Hui, F., Shokr, M., Granskog, M. A., Cheng, B., Vihma, T., and Cheng, X.: Winter Arctic Sea Ice Surface Form Drag During 1999-2021: Satellite Retrieval and Spatiotemporal Variability, IEEE Trans. Geosci. Remote Sensing, 1–1, https://doi.org/10.1109/TGRS.2023.3347694, 2024.

Zu, Y., Lu, P., Leppäranta, M., Cheng, B., and Li, Z.: On the Form Drag Coefficient Under Ridged Ice: Laboratory Experiments and Numerical Simulations From Ideal Scaling to Deep Water, Journal of Geophysical Research: Oceans, 126, e2020JC016976, https://doi.org/10.1029/2020JC016976, 2021.